# Natural variation in sugar tolerance associates with changes in signaling and mitochondrial ribosome biogenesis

Richard G Melvin[1,2†], Nicole Lamichane[1,2], Essi Havula[1,2‡], Krista Kokki[1,2], Charles Soeder[3], Corbin D Jones[3], Ville Hietakangas[1,2]*

[1]Faculty of Biological and Environmental Sciences, University of Helsinki, Helsinki, Finland; [2]Institute of Biotechnology, University of Helsinki, Helsinki, Finland; [3]Biology Department, The University of North Carolina at Chapel Hill, Carolina, United States

**\*For correspondence:**
ville.hietakangas@helsinki.fi

**Present address:** [†]University of Minnesota School of Medicine, Minnesota, United States; [‡]Charles Perkins Centre, University of Sydney, Sydney, Australia

**Competing interests:** The authors declare that no competing interests exist.

**Abstract** How dietary selection affects genome evolution to define the optimal range of nutrient intake is a poorly understood question with medical relevance. We have addressed this question by analyzing *Drosophila simulans* and *sechellia*, recently diverged species with differential diet choice. *D. sechellia* larvae, specialized to a nutrient scarce diet, did not survive on sugar-rich conditions, while the generalist species *D. simulans* was sugar tolerant. Sugar tolerance in *D. simulans* was a tradeoff for performance on low-energy diet and was associated with global reprogramming of metabolic gene expression. Hybridization and phenotype-based introgression revealed the genomic regions of *D. simulans* that were sufficient for sugar tolerance. These regions included genes that are involved in mitochondrial ribosome biogenesis and intracellular signaling, such as *PPP1R15/Gadd34* and *SERCA*, which contributed to sugar tolerance. In conclusion, genomic variation affecting genes involved in global metabolic control defines the optimal range for dietary macronutrient composition.
DOI: https://doi.org/10.7554/eLife.40841.001

## Introduction

Animals require macronutrients to sustain growth, reproduction and repair over their lifetimes and the balance between nutrients has been shown to have significant effects on development, reproduction and longevity (*Raubenheimer and Simpson, 1997*; *Lee et al., 2008*; *Arganda et al., 2017*). Most animals consume a variety of different foods to meet their nutritional needs (*Raubenheimer and Simpson, 1997*; *Lee et al., 2008*) and even closely related species make highly distinct diet choices (*Tinker et al., 2008*; *Goldman-Huertas et al., 2015*; *Salinas-Ramos et al., 2015*; *Costello et al., 2016*; *Han et al., 2016*). Therefore, it is conceivable that the impact of nutrient composition on various life history traits depends on the genetic makeup of the animal. Some closely related species are distinguished by variation in morphological structures that are specialized for obtaining nutrients from unique resources (trophic morphology) (*Malinsky et al., 2015*; *Parsons et al., 2016*; *Santana and Cheung, 2016*). Darwin's finches are the classic example of species that are differentiated in part by interspecific competition and specialization on an under-used food type (*De León et al., 2014*). Darwin's finches are considered 'imperfect dietary generalists' (*De León et al., 2014*) having similar preferred diets that overlap among species but specialize on a unique food when the preferred diet is limiting. Such difference in diet flexibility suggests that in addition to morphological differences, animals might also display differential metabolic flexibility, that is the capacity to adapt nutrient use to nutrient availability. While much recent attention has been paid to the genetics that underlie plasticity of trophic morphology in animals (*Malinsky et al.,*

**eLife digest** Animals meet their nutritional needs in a variety of ways. Some animals are specialists feeding only on one type of food; others are generalists that can choose many different kinds of food depending on the situation. Despite these differences in diet, animals have similar needs for basic cellular metabolism. This suggests that generalist and specialist species likely process the foods they eat in different ways in order to meet their basic needs. For example, the metabolism of generalist species may be more flexible to adapt to changing food sources.

To learn more about how metabolism evolves to respond to diet, scientists can study closely related species that eat different foods. For example, a species of fruit fly called *Drosophila simulans* is a generalist and its larvae can grow and develop by feeding on different kinds of decaying fruits and vegetables. Larvae of a closely related fruit fly called *Drosophila sechellia* are specialized to eat only the nutrient-poor *Morinda* fruit. Looking at how genetic differences between these species affect metabolism may provide scientists with clues about how these feeding strategies evolved.

Melvin et al. grew larvae of *D. sechellia* and *D. simulans* in different conditions. *D. sechellia* larvae thrived in low nutrient conditions, but died when exposed to high sugar foods. By contrast, *D. simulans* larvae tolerated high sugar levels, but did poorly in low-nutrient conditions.

Melvin et al. then bred the two species with each other, selecting flies that are genetically similar to *D. sechellia* but have the genes necessary for larvae to tolerate sugar. Analyzing the selected hybrid flies revealed genetic changes that explain the different survival abilities of each species. These changes suggest that *D. sechellia* rapidly evolved to thrive in low nutrient conditions, but the trade-off was losing their ability to tolerate high sugar levels.

Overall, the results presented by Melvin et al. suggest that genetic adaptions to food sources can occur quickly and drastically change metabolism. Further research will be needed to confirm if similar metabolic trade-offs developed as part of human evolution. If so, human populations that survived with limited nutrition for many generations may have a harder time adapting to high-sugar modern diets.

DOI: https://doi.org/10.7554/eLife.40841.002

*2015*; *Ledogar et al., 2016*; *McGirr and Martin, 2017*; *Burress et al., 2017*; *Zelditch et al., 2017*), less focus has been placed on metabolic regulators with regard to diet choice (*Turner and Thompson, 2013*). Metabolic phenotypes and diet tolerance is observed to vary with ecological diversification within and between species (*Matzkin et al., 2009*; *Reed et al., 2010*; *Matzkin et al., 2011*), and these phenotypic changes correlate with changes in gene expression (*Nazario-Yepiz et al., 2017*). However, it remains poorly understood, which genetic changes are causally important during evolution of diet choice and what kind of metabolic tradeoffs might emerge from adaptation to a new macronutrient composition.

Flexibility in the usage of metabolic pathways allows animals to accommodate changes in food nutrient content and availability. At the level of the organism, systemic nutrient levels are actively monitored by the so-called nutrient-sensing pathways composed of intra- and intercellular signaling pathways and gene regulatory networks, which ultimately control the activity of metabolic pathways (*Mattila and Hietakangas, 2017*). There are specific nutrient-sensing mechanisms for each type of macronutrient. For example, protein kinases mTOR complex 1 (mTORC1) and GCN2 respond to changes in amino acid availability (*Efeyan et al., 2015*), while the transcription factor complex Mondo/ChREBP-Mlx is activated in response to sugars (*Havula and Hietakangas, 2012*). In *Drosophila*, genetic mutations that impair nutrient-sensing pathways have revealed diet-specific phenotypes. Adult mutants of 4EBP, a target of mTORC1, are indistinguishable from controls when fed a protein-rich diet, but are sensitive to amino acid starvation (*Teleman et al., 2005*). On the other hand, *mlx* mutant larvae have impaired function of the sugar sensor Mondo/ChREBP-Mlx and grow normally when fed a low sugar diet (LSD), but are intolerant of high-sugar diet (HSD) (*Havula et al., 2013*). The *mlx* null mutants exhibit impaired growth, increased larval development time and reduced larvae to pupae survival when the high dietary sugar concentration is within the range available from natural food sources (*Havula et al., 2013*). Thus, nutrient-sensing pathways define the tolerated lower and upper limits of nutrient intake. These limits for each nutrient will further depend on

the availability of other nutrients. We call the inclusive matrix of tolerated macronutrient contents the 'macronutrient space'.

Here, we aimed to explore the natural variation of macronutrient space in closely related species. We hypothesized that the natural variation of diet choice as well as diet flexibility (specialist *vs.* generalist) is affected by genetically encoded differences that define the macronutrient space. To test this hypothesis, we studied two closely related *Drosophila* species that differ in diet choice, namely the generalist *D. simulans* and its specialist relative, *D. sechellia*. In nature, *D. simulans* larvae consume a range of decaying fruits that may contain high levels of sugars, whereas *D. sechellia* larvae grow on the unripe fruits of *Morinda citrifolia,* which has a low-sugar content (*Singh et al., 2012*). The two species occur together on islands of the Seychelles archipelago; however, *D. sechellia* adults and larvae are found infrequently on fruits other than that of *Morinda* (*R'Kha et al., 1991*; *Matute and Ayroles, 2014*). *D. simulans* and *D. sechellia* show strong dietary differentiation yet they are closely related and can form fertile female F1 hybrids (*Lachaise et al., 1986*). This makes the two species and their hybrids a tractable system for studying the genetics associated with determination of macronutrient space.

## Results

### Closely related *Drosophila* species have differential macronutrient spaces

Because the natural larval diet of the generalist species *D. simulans* may have a highly variable sugar content compared to that of the specialist species *D. sechellia*, we predicted that egg to pupa development time of these species would be dissociated along the sugar axis in a yeast × sugar macronutrient space. To test this prediction, we characterized larval development time and survival to pupa for both species across a macronutrient space consisting of a 5 × 5 grid of diets that varied in sucrose and yeast content. The species showed different phenotypes across the grid of diets (*Figure 1A*, *Table 1*). *D. simulans* larvae displayed rapid development and high larval survival on diets up to and including 20% sugar. In contrast, *D. sechellia* larvae displayed a more restricted space, with slowed development and reduced survival on high-protein diets containing >10% sugar and complete lethality on diet composed of 20% sucrose/20% yeast (*Figure 1A*, *Table 1*). For *D. sechellia* larvae, dietary sucrose concentration showed significant ($p < 0.001$) positive correlation with lengthened development time and significant ($p < 0.001$) negative correlation with survival, but such correlations were not observed for *D. simulans* larvae (*Table 1*). For both species, larval development time was negatively correlated and survival was positively correlated with dietary yeast concentration; however, the correlation was weaker for *D. sechellia* than for *D. simulans* (*Table 1*). To test the possibility that *D. simulans* and *D. sechellia* larvae differed by a behavioral feeding response to sugar, we assayed mouth hook extension rate for both species in the presence and absence of 20% sucrose (*Shen, 2012*; *Scheiner et al., 2014*). ANOVA showed no significant effect of species ($F_{(1, 12)} = 0.06$, $p = 0.81$), sugar concentration ($F_{(1, 12)} = 0.42$, $p = 0.53$), or their interaction ($F_{(1,12)} = 4.47$, $p = 0.06$) on feeding behavior (*Figure 1—figure supplement 1*).

Since nutrition affected both larval development time and survival similarly, we combined the data and calculated a so-called 'pupariation index' (Pupind) that takes both parameters into account. A high Pupind score is achieved with shorter larval development time and higher survival to pupal stage. Analysis of the Pupind of *D. simulans* and *sechellia* confirmed the poor performance of *sechellia* on high-sugar diets (*Figure 1A*, *Table 1*). We further analyzed our data by using a full generalized linear model (*glm*), which showed significant effects of genotype, percent sugar, percent yeast, and all interactions of the main effects on larval development time, larval survival, and Pupind (*Table 2*). However, while the effect of yeast on Pupind was stronger for *D. simulans* than for *D. sechellia* ($\omega^2 = 0.94$ and $0.33$, respectively), the effect of sugar on Pupind was substantially stronger for *D. sechellia* ($\omega^2 = 0.01$ and $0.35$ for *D. simulans* and *D. sechellia*, respectively) (*Table 3*). This further supports the conclusion that *D. sechellia* is sugar intolerant.

To confirm a genetic basis for the observed differential sugar tolerance between species, we assayed the larval development time and survival to pupa of *D. simulans* × *D. sechellia* F1 hybrid larvae across the 5 × 5 grid of diets. *D. simulans* and *D. sechellia* are closely related, having diverged from a common ancestor roughly 0.4 million years ago (*Kliman et al., 2000*), and hybrid females

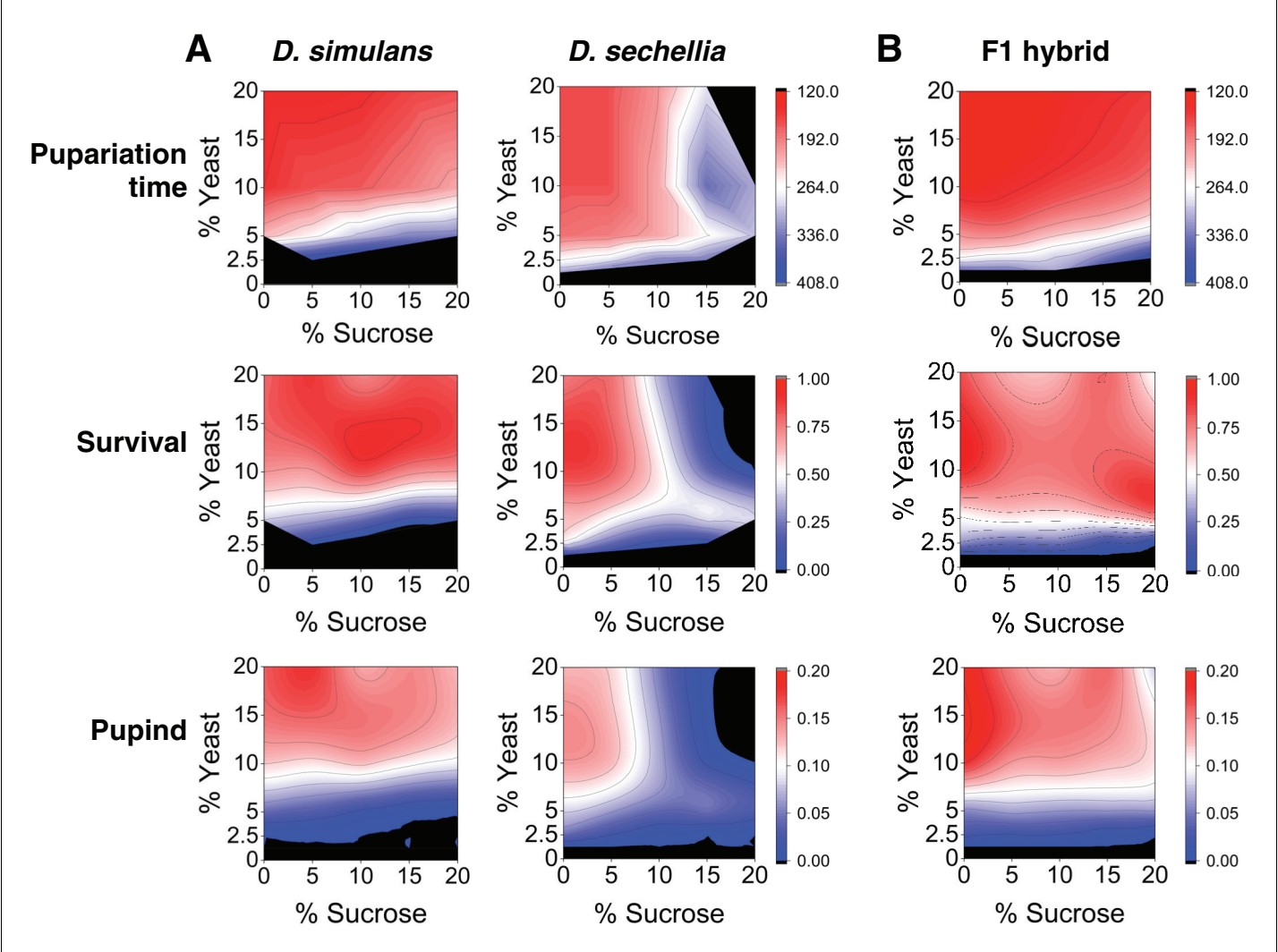

**Figure 1.** Differential macronutrient spaces of *Drosophila simulans* and *sechellia* with respect to sugar tolerance. (**A**) Larvae of *D. simulans* and *D. sechellia* showed differential pupariation time (h after egg-laying) and survival on high dietary sugar. Larval development was monitored on a 5 × 5 grid of varying yeast and sucrose levels. Pupariation index takes into account both survival and pupariation time. n = 5 replicates of 30 larvae/replicate for each genotype and diet. (**B**) Tolerance of high dietary carbohydrate was restored in the *D. sechellia* x *D. simulans* F1 hybrids. n = 5 replicates of 30 larvae/replicate for each diet.

DOI: https://doi.org/10.7554/eLife.40841.003

The following source data and figure supplement are available for figure 1:

**Source data 1.** Source data for *Figure 1A*.
DOI: https://doi.org/10.7554/eLife.40841.005
**Source data 2.** Source data for *Figure 1B*.
DOI: https://doi.org/10.7554/eLife.40841.006
**Figure supplement 1.** Feeding behavior did not differ significantly between the species.
DOI: https://doi.org/10.7554/eLife.40841.004

from the cross, *D. sechellia* male × *D. simulans* female, are fertile (*Lachaise et al., 1986*). We observed a clear rescue of sugar tolerant larval development and survival in the F1 hybrid (*Figure 1B*). This implied that the sugar intolerance phenotype of *D. sechellia* may be caused by altered function of genes underlying sugar tolerance

**Table 1.** Correlation analysis of nutrient space metrics.

| | Pearson correlation coefficient | |
| --- | --- | --- |
| | **% Yeast** | **% Sucrose** |
| Larval development time | | |
| *D. simulans* | −0.76*** | 0.21 |
| *D. sechellia* | −0.45*** | 0.59*** |
| F₁ hybrid | −0.82*** | 0.12 |
| Larval survival | | |
| *D. simulans* | 0.88*** | −0.08 |
| *D. sechellia* | 0.36*** | −0.58*** |
| F₁ hybrid | 0.69*** | −0.09 |
| Pupariation Index | | |
| *D. simulans* | 0.93*** | −0.12 |
| *D. sechellia* | 0.42*** | −0.69*** |
| F₁ hybrid | 0.76*** | −0.19* |

$*P < 0.05$; $**P < 0.01$, $***P < 0.001$

DOI: https://doi.org/10.7554/eLife.40841.007

**Table 2.** Generalized linear model (GLM) details for pupariation index (Pupind), larval survival, and development time.

Models assumed a normal distribution and used an identity link function. Error d.f. = 300 for all comparisons.

| Trait | Effect | d.f. | Log ratio $\chi^2$ | P |
| --- | --- | --- | --- | --- |
| Pupind | Genotype | 2 | 297.84 | <0.001*** |
| | Sugar | 4 | 383.41 | <0.001*** |
| | Yeast | 4 | 1064.98 | <0.001*** |
| | Genotype × Sugar | 8 | 177.07 | <0.001*** |
| | Genotype × Yeast | 8 | 489.76 | <0.001*** |
| | Sugar × Yeast | 16 | 286.75 | <0.001*** |
| | Genotype × Sugar × Yeast | 32 | 218.40 | <0.001*** |
| Survival | Genotype | 2 | 205.79 | <0.001*** |
| | Sugar | 4 | 270.76 | <0.001*** |
| | Yeast | 4 | 971.02 | <0.001*** |
| | Genotype × Sugar | 8 | 240.94 | <0.001*** |
| | Genotype × Yeast | 8 | 416.18 | <0.001*** |
| | Sugar × Yeast | 16 | 135.45 | <0.001*** |
| | Genotype × Sugar × Yeast | 32 | 303.48 | <0.001*** |
| Dev. time | Genotype | 2 | 96.40 | <0.001*** |
| | Sugar | 4 | 109.69 | <0.001*** |
| | Yeast | 4 | 476.02 | <0.001*** |
| | Genotype × Sugar | 8 | 53.95 | <0.001*** |
| | Genotype × Yeast | 8 | 150.31 | <0.001*** |
| | Sugar × Yeast | 16 | 30.67 | <0.05 * |
| | Genotype × Sugar × Yeast | 32 | 125.34 | <0.001*** |

DOI: https://doi.org/10.7554/eLife.40841.008

**Table 3.** Effect sizes by trait and genotype.

| Trait | Genotype | Effect | -log(*P*) | $\varpi^2$ |
|---|---|---|---|---|
| Pupind | *D. simulans* | Sugar | 12.79 | 0.01 |
| | | Yeast | 99.04 | 0.94 |
| | | Sugar × Yeast | 9.43 | 0.02 |
| | *D. sechellia* | Sugar | 43.52 | 0.35 |
| | | Yeast | 41.83 | 0.33 |
| | | Sugar × Yeast | 26.61 | 0.22 |
| | F1 hybrid | Sugar | 18.62 | 0.04 |
| | | Yeast | 83.58 | 0.86 |
| | | Sugar × Yeast | 16.73 | 0.85 |
| Larval survival | *D. simulans* | Sugar | 5.15 | 0.01 |
| | | Yeast | 95.39 | 0.93 |
| | | Sugar × Yeast | 13.71 | 0.03 |
| | *D. sechellia* | Sugar | 40.15 | 0.34 |
| | | Yeast | 41.58 | 0.36 |
| | | Sugar × Yeast | 22.47 | 0.19 |
| | F1 hybrid | Sugar | 4.15 | 0.01 |
| | | Yeast | 76.78 | 0.88 |
| | | Sugar × Yeast | 11.76 | 0.05 |
| Dev. time | *D. simulans* | Sugar | 3.53 | 0.01 |
| | | Yeast | 75.52 | 0.90 |
| | | Sugar × Yeast | 7.00 | 0.03 |
| | *D. sechellia* | Sugar | 11.34 | 0.18 |
| | | Yeast | 18.28 | 0.34 |
| | | Sugar × Yeast | 3.78 | 0.09 |
| | F1 hybrid | Sugar | 5.58 | 0.05 |
| | | Yeast | 36.39 | 0.62 |
| | | Sugar × Yeast | 5.80 | 0.08 |

DOI: https://doi.org/10.7554/eLife.40841.009

## Introgression of sugar tolerance phenotype

To generate flies having the minimal *D. simulans* genomic regions essential for sugar tolerance in a mostly *D. sechellia* genomic background, we sought to introgress the sugar tolerance phenotype from *D. simulans* into a mostly *D. sechellia* genetic background. To do this, we used the phenotype-based introgression approach of Earley and Jones (*Earley and Jones, 2011*) (*Figure 2A*). Dietary sugar content of 20% provided a strong selection, since no survivors of the *D. sechellia* parental line were observed in these conditions. After 10 generations of backcrossing with selection, we observed that tolerance of an HSD (20% yeast/20% sugar) in the backcross larvae was equal to that of *D. simulans* (*Figure 2B*). A control fly line that was backcrossed in the same manner, but not selected on a high-sugar diet, showed only minimal tolerance for HSD (*Figure 2B*). Since the introgression was performed by repeated backcrossing of *D. sechellia* males with the hybrid line (maternally *D. simulans*), the mitochondrial genomes of the selected and control lines are the same. Therefore, the observed phenotypic differences are due to the nuclear genome. Morphologically, the introgressed lines resemble *D. sechellia*, including genital arch morphology (data not shown). Metabolic analysis of the parental and introgressed lines revealed that the sugar intolerant *D. sechellia* and no-selection control lines were less efficient in clearing glucose from circulation after challenge with high-sugar diet (*Figure 2C*). This suggests that the pathways controlling energy metabolism or their response toward high-sugar diet are affected by the genomic regions underlying sugar tolerance.

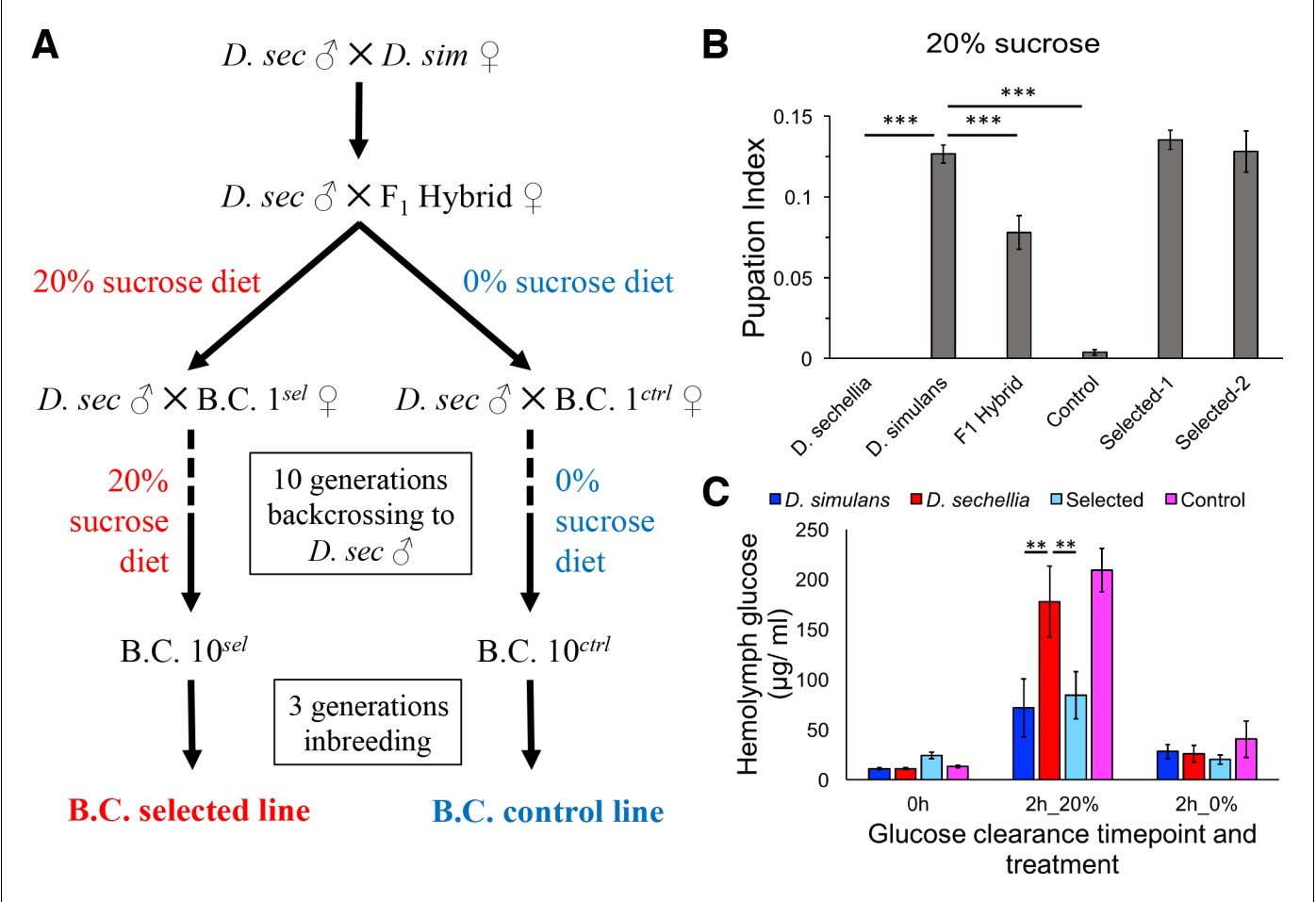

**Figure 2.** Introgression of *D. simulans* sugar tolerance into *D. sechellia* genome through repeated backcrosses on selective diet. (**A**) Construction of the sugar selected and control backcross (B.C.) lines through phenotype-based introgression. Dietary sugar content of 20% provided a strong selection, since no survivors of the *D. sechellia* parental line were observed in these conditions. (**B**) Sugar tolerance of selected lines was similar to that in the parental *D. simulans* line, while the sugar tolerance of the control line resembled to that of *D. sechellia*. Error bars display standard error of the mean. n = 5 replicates of 30 larvae/replicate for each genotype. Dunnett's test (|d| = 2.70, α = 0.05) showed that *D. sechellia* and the control backcross line had significantly reduced sugar tolerance compared to *D. simulans* while sugar tolerance of the two HSD-selected backcross lines did not differ from that of *D. simulans*. (**C**) The sugar intolerant control line showed impaired clearance of hemolymph glucose, similar to *D. sechellia*. Hemolymph glucose was measured from larvae on LSD, after 2 hr on HSD, and after 2 hr of transferring of HSD-fed larvae back to LSD. Error bars display standard error of the mean. n = 5 replicates of 10 larvae/replicate for each genotype and diet. Dunnett's test (|d| = 2.62, α = 0.05) showed that after feeding for 2 hr on HSD, *D. sechellia* and the control backcross line had significantly elevated hemolymph glucose compared to that of *D. simulans* while that of the selected line did not differ from the *D. simulans* level. **p < 0.01, ***p < 0.001.

DOI: https://doi.org/10.7554/eLife.40841.010

## Larval gene expression profiles are strongly associated with sugar tolerance

In order to achieve a genome-wide view of the gene expression profiles in the sugar tolerant and intolerant lines, we used RNAseq analysis and assayed 3rd instar larvae fed continuously on LSD (20% yeast) as well as following acute exposure to HSD (20% yeast/20% sugar for 8 hr) (*Figure 3A*). Global comparison of the gene expression by sample clustering revealed striking association between expression profiles and sugar tolerance. The gene expression profile of the sugar-selected hybrid had high similarity with that of *D. simulans*, while the sugar intolerant control hybrid clustered close to *D. sechellia* (*Figure 3B*). This implies, surprisingly, that the genetic factors underlying the differences in sugar tolerance explain the majority of the differential gene expression between the parental species. We further identified the genes that were differentially expressed in the tolerant

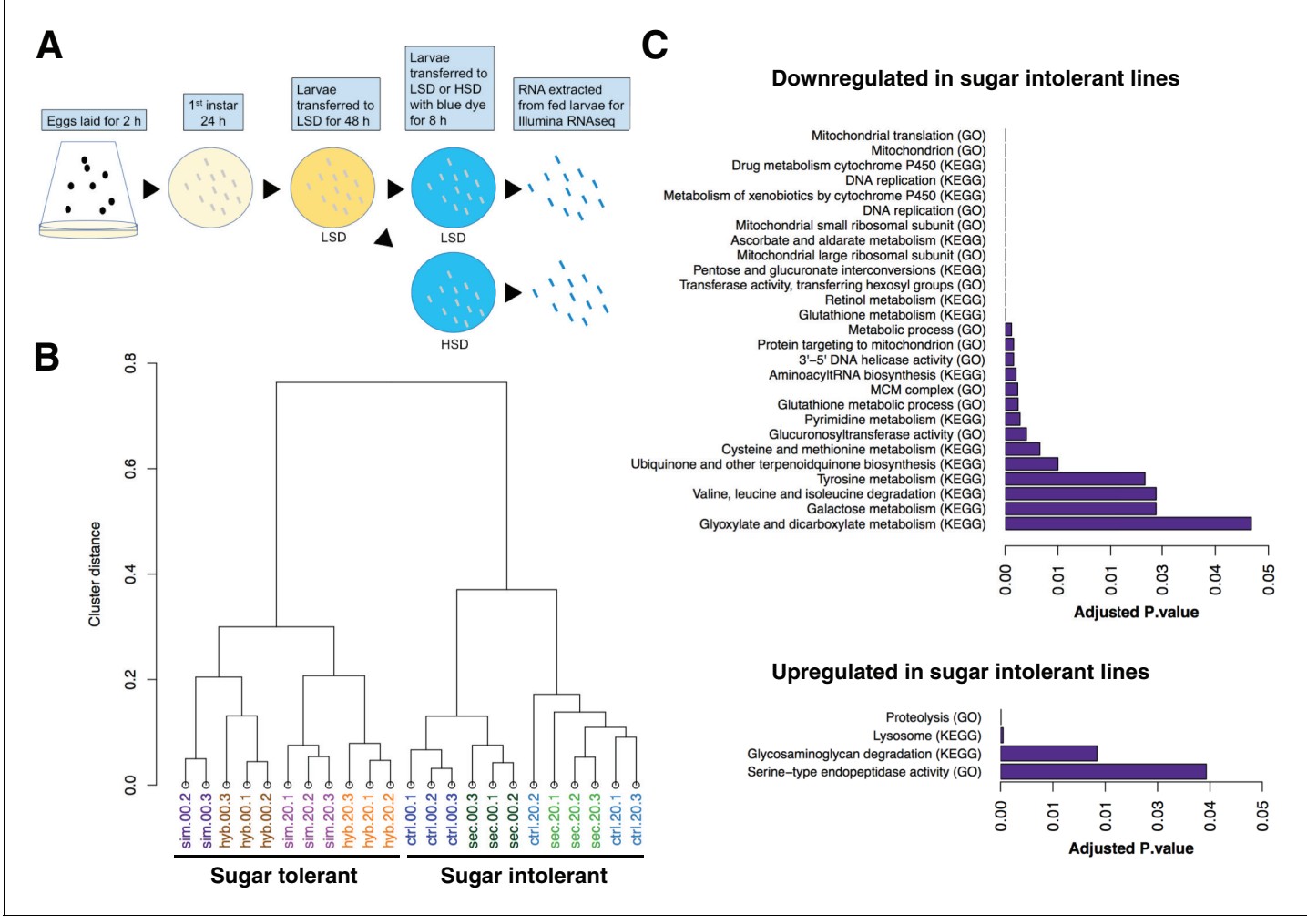

**Figure 3.** Global gene expression changes associated with sugar tolerance. (**A**) Schematic representation of RNAseq sample preparation. Parental lines and backcrossed hybrid lines were fed on LSD or transferred acutely (8 hr) on HSD, followed by RNA extraction and RNA sequencing. (**B**) Sample clustering reveals tight association between global gene expression profiles and sugar tolerance. Sample clustering was based on Pearson correlation and it was performed using R/Bioconductor package pvclust. Correlation was used as distance matrix. (**C**) Summary of selected functional groups significantly enriched among genes displaying differential expression in sugar tolerant *vs.* intolerant lines.

DOI: https://doi.org/10.7554/eLife.40841.011

*vs.* intolerant genotypes, focusing on genes that differed significantly ($p < 0.05$) when both sugar tolerant genotypes were compared to both sugar intolerant genotypes on HSD. Genes with reduced expression in both sugar intolerant lines displayed significant ($p < 0.05$) enrichment in functional categories related to mitochondrial ribosome, detoxification (e.g. cytochrome P450 and glutathione metabolism), growth control (ribosome biogenesis), carbohydrate metabolism (starch and sucrose metabolism) as well as several categories related to amino acid metabolism (*Figure 3C*). On the other hand, genes with high expression in sugar intolerant lines displayed overrepresentation of proteolysis and lysosome (*Figure 3C*).

We have earlier observed that the *mlx*[1] null mutant larvae, lacking functional sugar sensing by Mondo-Mlx, display strong sugar intolerance, similar to *D. sechellia* (*Havula et al., 2013*). To test if the sugar intolerant *D. sechellia* lines show similarities to *mlx*[1] mutants in gene regulation, we compared the *simulans/sechellia* RNAseq dataset with that of *mlx*[1] null mutant, published earlier (*Mattila et al., 2015*). There was significant similarity between the gene expression profiles of the sugar intolerant genotypes (*Figure 4A*). For example, 30% (174/587; $p = 1.2 \times 10^{-69}$) of the genes downregulated in *mlx*[1] mutants displayed reduced expression in *D. sechellia* and the control hybrid

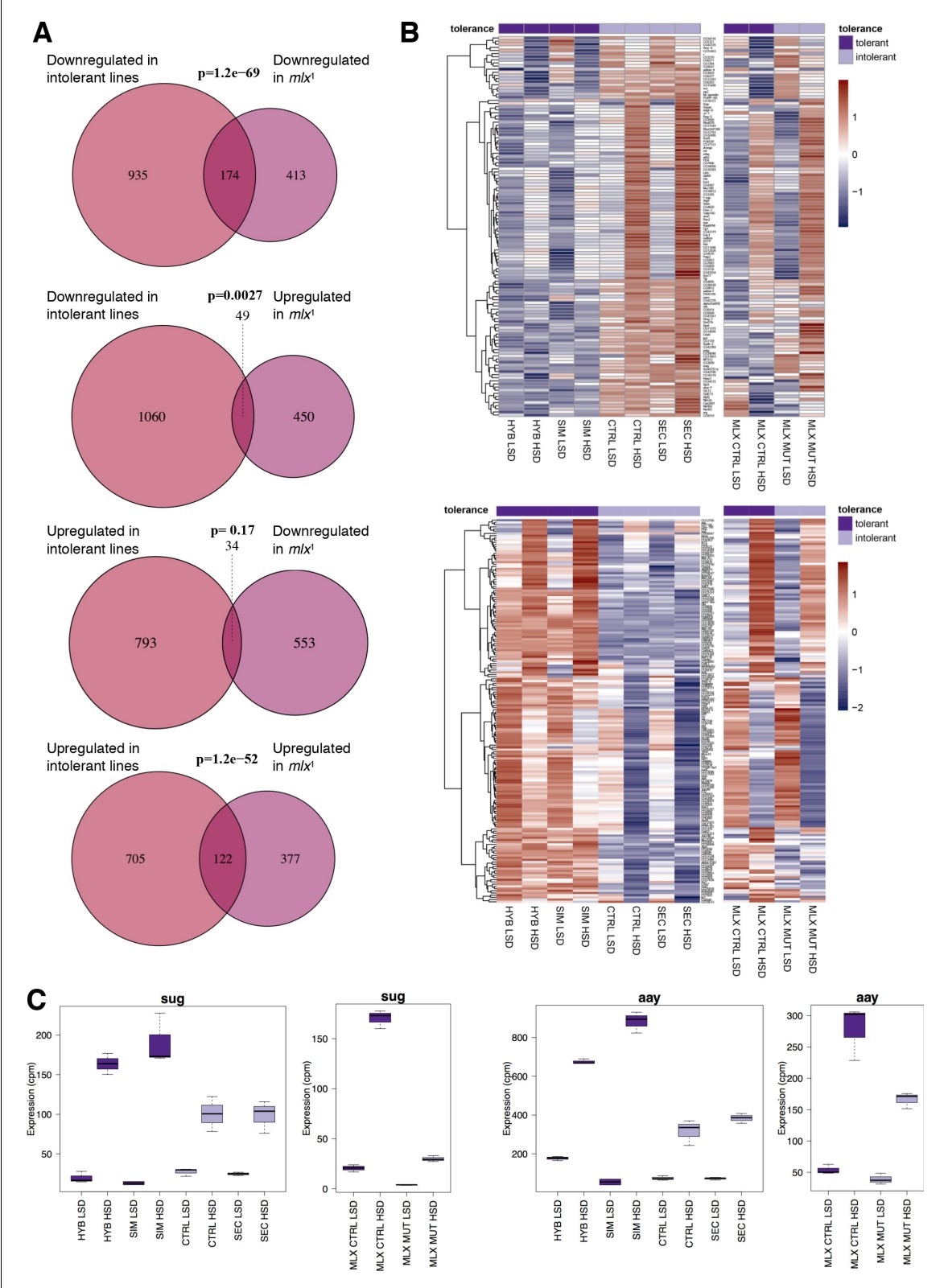

**Figure 4.** Significant overlap between gene expression profiles of sugar intolerant lines and *mlx* mutants. (**A**) Comparison of genes differentially expressed in sugar tolerant *vs.* intolerant lines with Mlx target genes. Gene expression profiles associated with sugar intolerance show highly significant overlap with profiles of *mlx*[1] mutants. (**B**) Heat maps of the overlapping gene sets show similarities in gene sugar responsiveness in sugar intolerant

*Figure 4 continued on next page*

*Figure 4 continued*
lines and *mlx*[1] mutants. Sugar tolerance/intolerance phenotypes of the analyzed lines are indicated in color. (**C**) Known sugar tolerance genes *sugarbabe* (*sug*) and *astray* (*aay*) show weaker sugar induction in sugar intolerant lines, resembling *mlx*[1] mutants.
DOI: https://doi.org/10.7554/eLife.40841.012

line (*Figure 4A*). Global comparison of all genes with similar gene expression differences in tolerant and intolerant lines revealed a high degree of similarity in the gene expression patterns (*Figure 4B*). These overlapping gene sets included several genes that were upregulated upon sugar feeding in sugar tolerant genotypes, but the activation was either absent or reduced in the sugar intolerant genotypes (*Figure 4B*). These include, for example the transcription factor *sugarbabe* and Phospho-serine phosphatase *astray* (*Figure 4C,D*), which we have earlier shown to be essential for sugar toler-ance (*Mattila et al., 2015*).

## Sugar tolerance associated with introgression of chromosome 2R

To determine the introgressed genomic regions and genes associated with the sugar tolerant phe-notype, we sequenced the whole genomes of the two parental, sugar-selected and non-selected control fly lines and identified species-specific SNPs across the genome. SNP analysis showed three small and one large region of *D. simulans* SNPs on chromosome arm 2R, while all other regions of the genome showed an almost completely *D. sechellia* SNP signature (*Figure 5A and B*). Locations of the introgressions relative to nucleotide positions on the *D. melanogaster* chromosome arm 2R were from approximately 5,758,067 to 6,085,625 (spanning 25 annotated *D. melanogaster* genes); from 6,600,044 to 6,810,530 (spanning 35 annotated genes); and 21,774,876 to 24,092,079 (span-ning 312 annotated genes) (*Supplementary file 1*). Majority of the introgressed genes showed no significant changes in gene expression (*Figure 5C*). In total, 40 introgressed genes were significantly ($p < 0.05$) downregulated in the intolerant lines, while 24 displayed elevated expression associated with sugar intolerance (*Figure 5C*; *Supplementary file 1*). This suggests that the global gene expression differences between sugar tolerant and intolerant lines are likely due to a small number of loci, which control the expression of a large number of downstream genes.

## Genes involved in mitochondrial ribosome biogenesis and intracellular signaling contribute to sugar tolerance

To identify genes from the introgressed chromosome 2R regions that were potentially responsible for the sugar tolerance phenotype, we utilized the genetic toolkit of *D. melanogaster*, a dietary gen-eralist and close relative of *D. simulans* and *sechellia* with sugar tolerance similar to that of *D. simu-lans* (*Figure 6—figure supplement 1*). From the total number of 372 introgressed genes, we selected 102 genes based on their annotation, with a putative metabolic or regulatory function to be screened for survival on low (20% yeast) and high (20% yeast/20% sugar) sugar diets (*Supplementary file 1*).

The screen identified several genes with a sugar intolerant phenotype. Interestingly, three of the identified sugar tolerance genes, *mRpL43*, *CG4882* and *bonsai* encode components of the mito-chondrial ribosome (*Figure 6A–C*). Furthermore, all of them displayed reduced expression in sugar intolerant genotypes (*Figure 6D–F*), implying that reduced capacity mitochondrial protein biosyn-thesis contributes to the sugar intolerance phenotypes. In addition to mitochondrial genes, our screen identified several sugar tolerance genes with a role in signaling. RNAi knockdown of *Sarco/endoplasmic reticulum Ca²⁺-ATPase* (*SERCA*), *Protein phosphatase 1 regulatory subunit 15* (*PPP1R15, Gadd34*), or *Phosphotidylinositol 3 kinase 59F* (*Pi3K59F*) led to strongly impaired larval growth on high-sugar diet, with only a few larvae surviving to pupae (*Figure 7A–C*). Furthermore, the expression of *SERCA* was downregulated in *D. sechellia* and in the sugar intolerant control line (*Figure 7D–F*). All these three genes have been linked to metabolic processes. *SERCA* pumps $Ca^{2+}$ into endoplasmic reticulum (ER) and is involved in control of lipid homeostasis (*Bi et al., 2014*), while *Pi3K59F* is a known regulator of autophagy (*Juhász et al., 2008*). *PPP1R15/Gadd34* is best known for its function as a negative regulator of the integrated stress response pathway, including amino acid sensing kinase GCN2 (*Malzer et al., 2013*). Furthermore, we found three additional genes

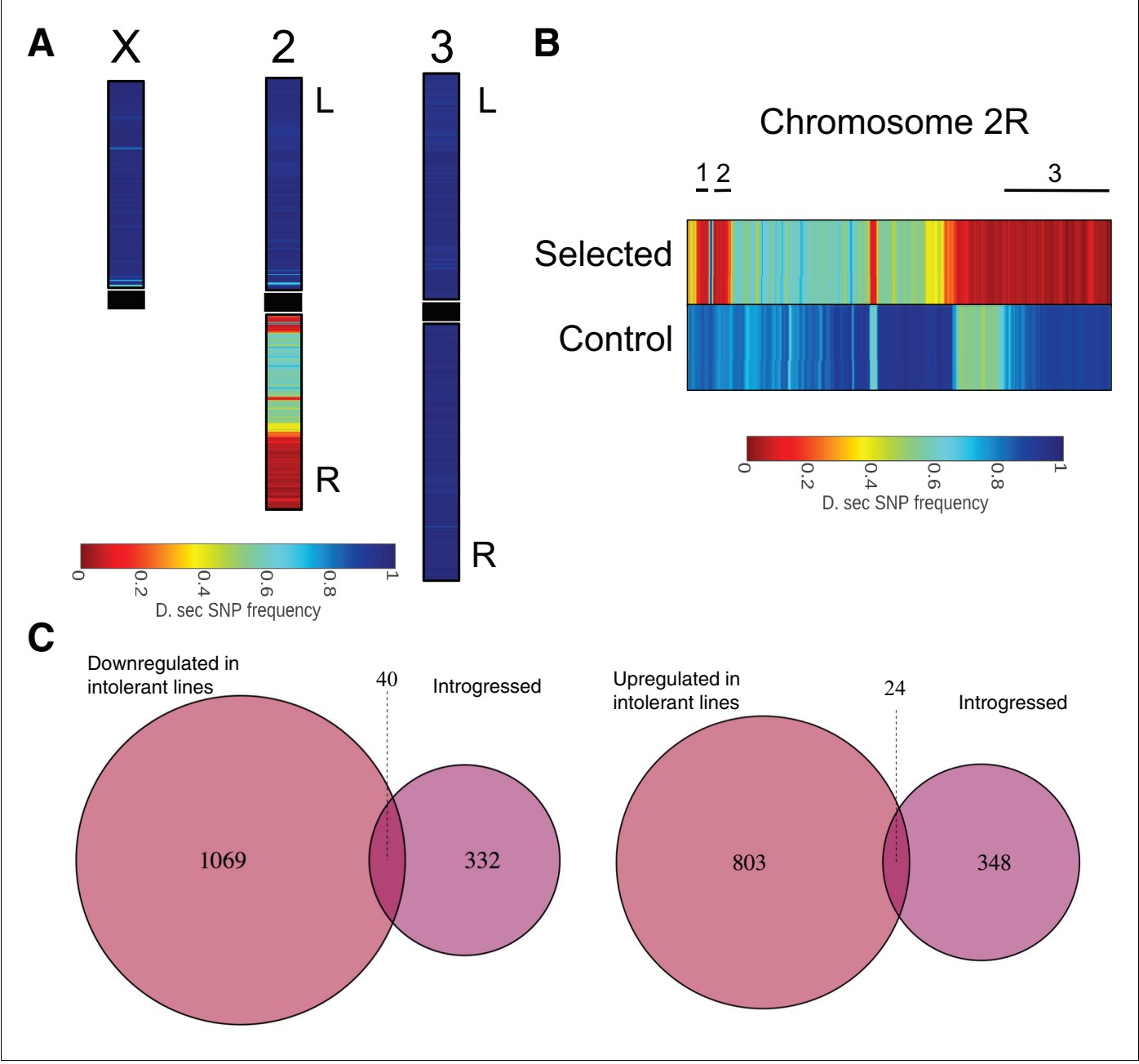

**Figure 5.** *D. simulans* SNP signature was introgressed into a mostly *D. sechellia* SNP signature background. (**A**) Color shows the frequency distribution of *D. simulans*-specific SNPs displayed along the chromosomes. (**B**) Frequency distribution of *D. simulans*-specific SNPs displayed on chromosome arm 2R for the sugar selected (top) and not-selected control (bottom) backcross lines. Black lines above the heat maps indicate the three sugar tolerance-associated introgressed regions. (**C**) Limited overlap between introgressed genes and genes that are either up- or downregulated in sugar tolerant lines.

DOI: https://doi.org/10.7554/eLife.40841.013

(*Taldo*, *Dpit47*, *GlcT-1*) displaying milder phenotypes, namely reduced eclosion on high-sugar diet (*Figure 7—figure supplement 1*).

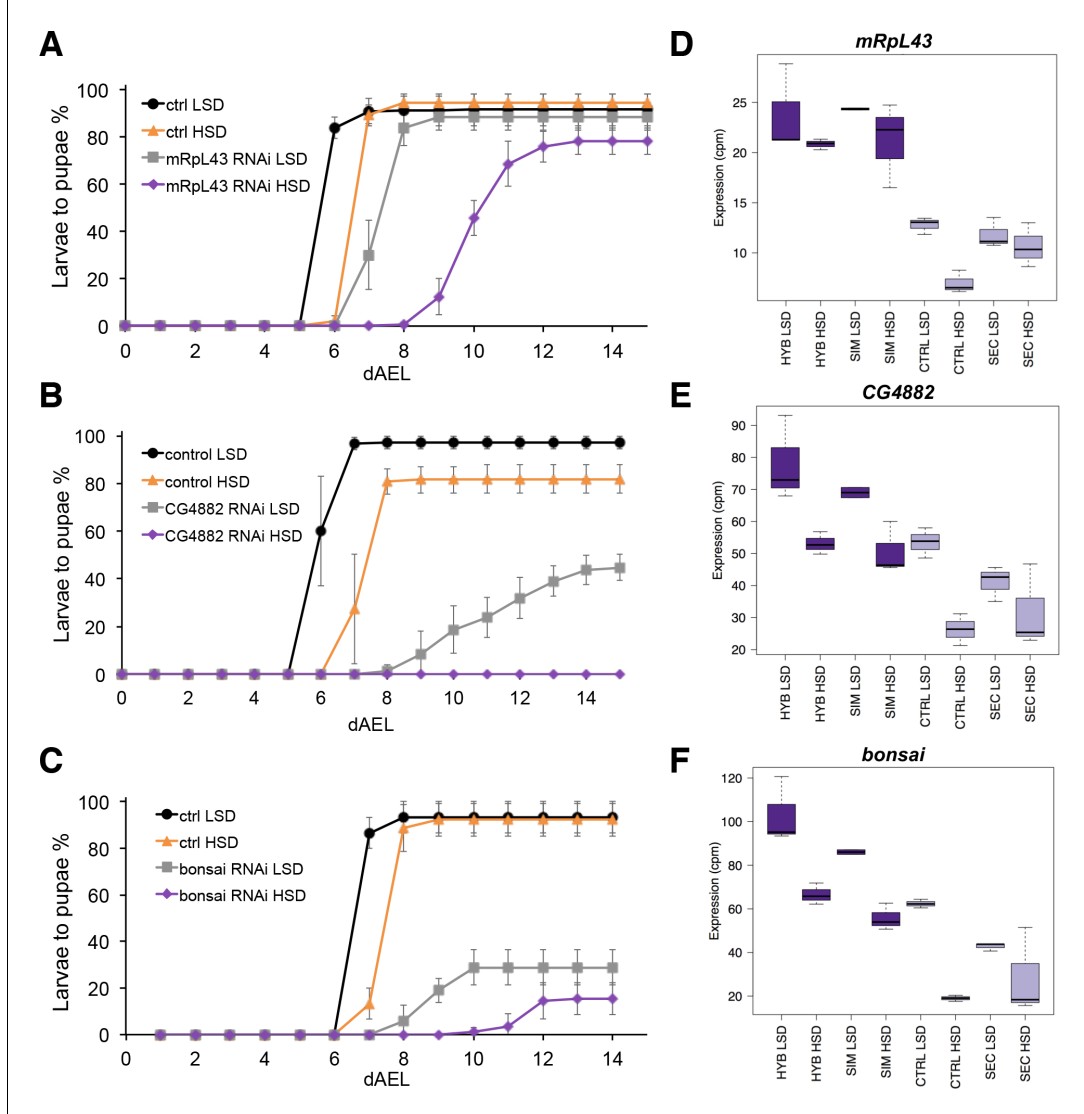

**Figure 6.** Low expression of mitochondrial ribosome genes contributes to sugar intolerance. (**A**) Pupariation kinetics of control and *mRpL43* RNAi larvae (Ubi-GAL4>), n = 7 replicates of 30 larvae/replicate for each genotype and diet. Error bars display standard deviation. (**B**) Pupariation kinetics of control and CG4882 RNAi larvae (Ubi-GAL4>), n = 8 replicates of 30 larvae/replicate for each genotype and diet. Error bars display standard deviation. (**C**) Pupariation kinetics of control and *bonsai* RNAi larvae (Fb-GAL4>), n = 3 replicates of 30 larvae/replicate for each genotype and diet. Error bars display standard deviation. (**D–F**) Relative expression of *mRpL43*, *CG2882*, and *bonsai* genes in sugar tolerant (hybrid and *D. simulans*) and intolerant (ctrl and *D. sechellia*) lines on low- and high-sugar diets identified by RNAseq. dAEL: days after egg laying.
DOI: https://doi.org/10.7554/eLife.40841.014

The following source data and figure supplement are available for figure 6:

**Source data 1.** Source data for *Figure 6A*.
DOI: https://doi.org/10.7554/eLife.40841.016
**Source data 2.** Source data for *Figure 6B*.
DOI: https://doi.org/10.7554/eLife.40841.017
**Source data 3.** Source data for *Figure 6C*.
DOI: https://doi.org/10.7554/eLife.40841.018
**Figure supplement 1.** Macronutrient space of *Drosophila melanogaster* shows high sugar tolerance.
DOI: https://doi.org/10.7554/eLife.40841.015

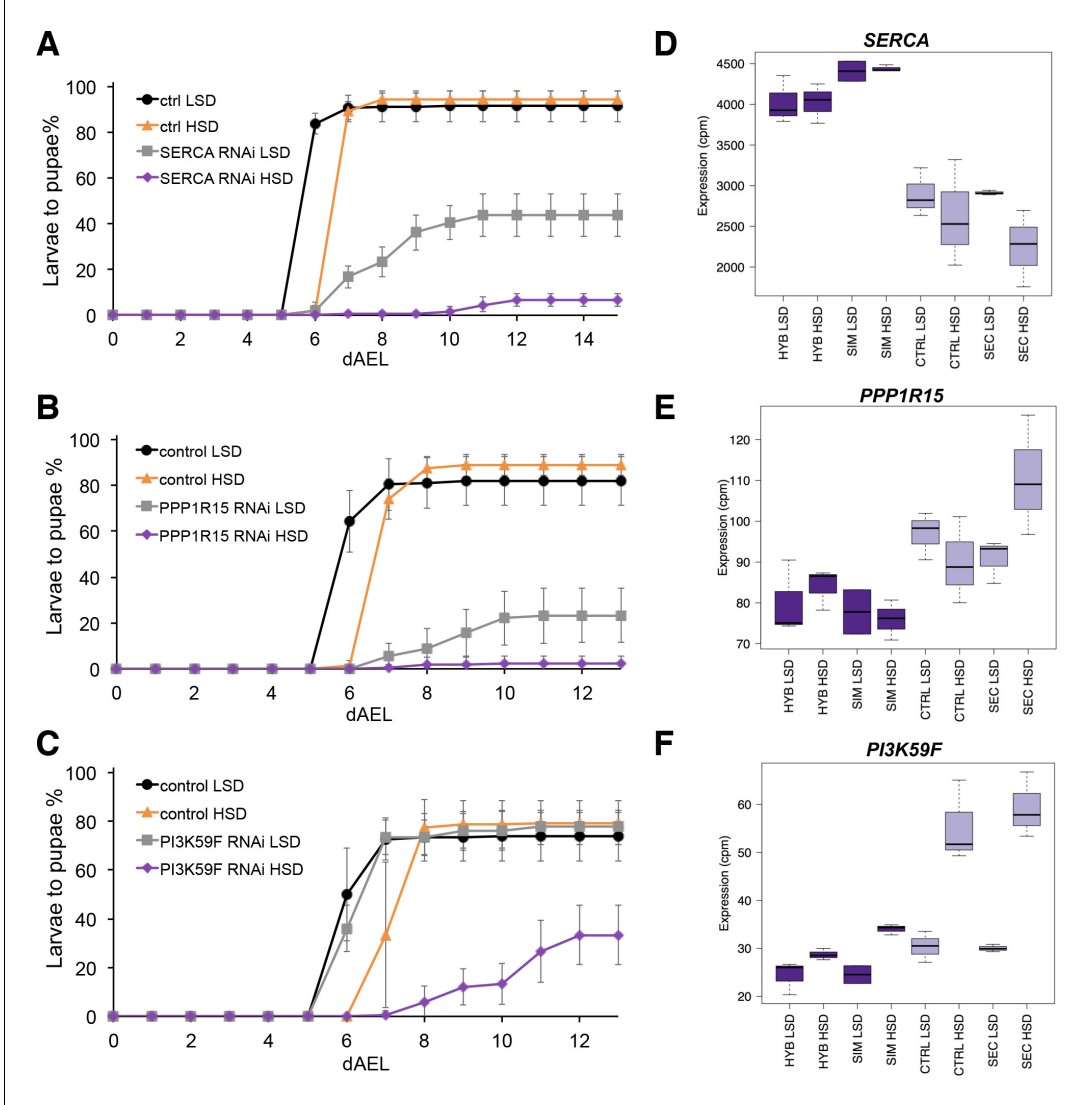

**Figure 7.** Several genes involved in signaling influence sugar tolerance. (A) Pupariation kinetics of control and *SERCA* RNAi larvae (Ubi-GAL4>), n = 7 replicates of 30 larvae/replicate for each genotype and diet. Error bars display standard deviation. (B) Pupariation kinetics of control and *PPP1R15* RNAi larvae (Ubi-GAL4>), n = 13 replicates of 30 larvae/replicate for each genotype and diet. Error bars display standard deviation. (C) Pupariation kinetics of control and *Pi3K59F* RNAi larvae (Tub-GAL4>), n = 5 replicates of 30 larvae/replicate for each genotype and diet. Error bars display standard deviation. (D–F) Relative expression of *SERCA*, *PPP1R15*, and *PI3K59F* genes in sugar tolerant (hybrid and *D. simulans*) and intolerant (ctrl and *D. sechellia*) lines on low- and high-sugar diets identified by RNAseq. dAEL: days after egg laying.

DOI: https://doi.org/10.7554/eLife.40841.019

The following source data and figure supplement are available for figure 7:

**Source data 1.** Source data for *Figure 7A*.
DOI: https://doi.org/10.7554/eLife.40841.021
**Source data 2.** Source data for *Figure 7B*.
DOI: https://doi.org/10.7554/eLife.40841.022
**Source data 3.** Source data for *Figure 7C*.
DOI: https://doi.org/10.7554/eLife.40841.023
**Figure supplement 1.** Genes with a modest impact on sugar tolerance.
DOI: https://doi.org/10.7554/eLife.40841.020

## Genomic changes in *SERCA* promoter lead to differential gene expression

Next we wanted to assess the level of genomic variation in the candidate genes identified, first focusing on the four genes (*mRpL43*, *CG4882*, *bonsai*, and *SERCA*) that displayed reduced expression in sugar intolerant lines. We mapped the density of nucleotide differences between *D. simulans* and *sechellia* using a sliding window of 100 bases within these specific gene regions. While all genomic regions displayed areas of high SNP density, the promoter region of the *SERCA* gene was found to be particularly variable (*Figure 8A*; *Figure 8—figure supplement 1*). To validate the functional impact of this variation, we cloned 1.2 kB fragments with putative promoter regions of *D. simulans* and *sechellia SERCA* gene in front of a *lacZ* reporter and generated in vivo reporter lines in *D. melanogaster* (*Figure 8A*). Indeed, the *D. sechellia*-derived promoter displayed significantly (p < 0.01) lower activity than the respective region of *D. simulans*, confirming the functional importance of the SNPs in the *SERCA* promoter (*Figure 8B*).

## High degree of amino acid changing coding region variation in the *PPP1R15* gene

We also looked for potential coding region changes in the candidate genes. For all of the hits, the *D. sechellia* DNA sequence contained nucleotide substitutions that cause amino acid differences in the encoded protein as compared to *D. simulans* (*Table 4*). Most genes in the set of hits had substantially higher number of silent than amino acid altering nucleotide differences, which implies purifying selection. In contrast, there were 10 amino acid changing and only five silent nucleotide differences in the *PPP1R15* sequence of *D. sechellia* compared to that of *D. simulans*. The rate of amino acid changing to silent mutations ($K_a/K_s$) in *PPP1R15* was 0.58 indicating reduced purifying selection along the *D. sechellia* lineage (*Table 4*). Plausible alternative explanations for the higher $K_a/K_s$ include the introduction of a new selective pressure on the founder population of *D. sechellia*.

## Trade-off of sugar tolerance with survival in a low nutrient environment

Low sugar tolerance in *D. sechellia* could be due to genetic drift in a low sugar dietary environment lacking selection or may be caused by a trade-off for an altered function that provides *D. sechellia* with selective advantage. To test if sugar tolerance is associated with *Morinda* toxin tolerance, we

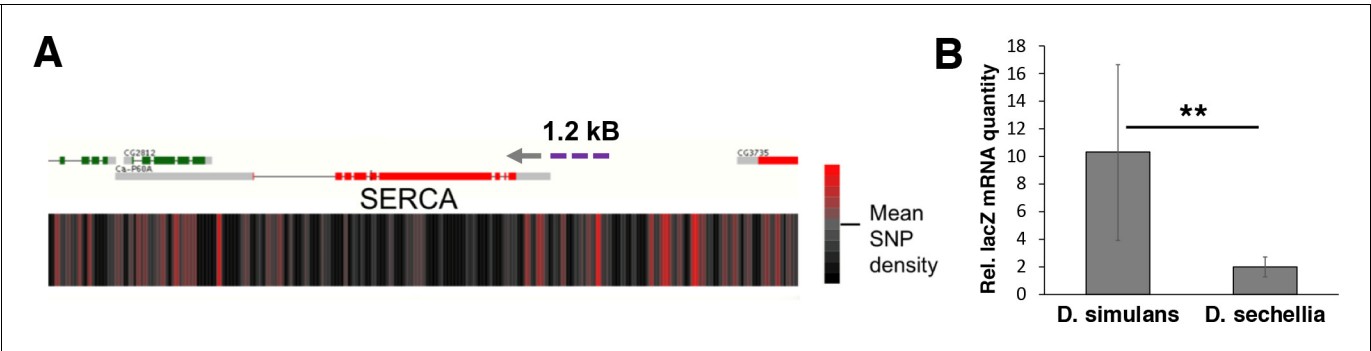

**Figure 8.** Genomic variation of SERCA promoter leads to differential promoter activity. (**A**) SNP density maps comparing *D. sechellia* to *D. simulans* on regions surrounding the *SERCA* (*Ca-P60A*) gene. mRNA transcript models for each gene region are shown above SNP density heat maps with green and red representing coding regions on the (+) and (-) strand, respectively, and grey indicating non-coding regions. Direction of transcription is also indicated with a grey arrowhead. Heatmaps represent the density of SNP differences between *D. sechellia* and *D. simulans* in overlapping windows of 100 nt slid in 25 nt increments along the region. The promoter fragment cloned into the in vivo reporter is indicated as violet dashed line. (**B**) Relative mRNA (qPCR) expression of *lacZ* reporter gene downstream of 1.2 kB fragments of *D. simulans* and *D. sechellia SERCA* promoters reveals lower activity of *D. sechellia*-derived promoter. n = 8 replicates of 8 larvae/replicate for each genotype and diet. Error bars display standard deviation. **p < 0.01 (student's t-test).

DOI: https://doi.org/10.7554/eLife.40841.024

The following figure supplement is available for figure 8:

**Figure supplement 1.** Genomic variation in the mitochondrial ribosome encoding gene regions.
DOI: https://doi.org/10.7554/eLife.40841.025

**Table 4.** Nucleotide substitution differences between *D. simulans* and *D. sechellia* in genes identified as hits in the *D. melanogaster* RNAi screen.
AA: amino acid.

| Gene | AA changing | Silent | $k_a/k_s$ |
|---|---|---|---|
| *PPP1R15* (CG3825) | 10 | 5 | 0.58 |
| *Pi3K59F* (CG5373) | 2 | 23 | 0.03 |
| CG4882 | 4 | 8 | 0.15 |
| *Taldo* (CG2827) | 1 | 6 | 0.05 |
| *Dpit47* (CG3189) | 4 | 9 | 0.13 |
| *GlcT-1* (CG6437) | 3 | 10 | 0.01 |
| *bonsai* (CG4207) | 2 | 8 | 0.13 |
| *mRPL43* (CG5479) | 1 | 13 | 0.03 |
| *SERCA* (CG3725) | 6 | 16 | 0.13 |

DOI: https://doi.org/10.7554/eLife.40841.026

selected hybrid larvae on *Morinda* toxin and performed three generations of selection. Selection for tolerance of the *Morinda* toxins had no significant impact on sugar tolerance (***Figure 9—figure supplement 1***). Moreover, the sugar tolerant introgression lines were not sensitive to *Morinda* toxins (***Figure 9—figure supplement 1***), further implying that toxin tolerance is genetically independent of sugar tolerance.

Since no association between sugar tolerance and toxin tolerance was found, we hypothesized that poor sugar tolerance is associated with improved fitness in low-sugar nutrient space. Therefore, we determined regions of the diet space where *D. sechellia* larvae would have an advantage compared to those of *D. simulans*. We subtracted the pupariation rate of *D. simulans* from that of *D. sechellia* and plotted the difference across the diet space. The subtracted diet space surface shows that *D. sechellia* and *D. simulans* larvae have clearly separated peaks where they hold an advantage (***Figure 9A***). The strongest advantage for *D. sechellia* was observed when the yeast content was 5% or lower and sugar levels were close to zero. In order to test, whether the high fitness of *D. sechellia* on low energy diet is a tradeoff for sugar tolerance, we raised the sugar selected and control introgression lines on 2.5% yeast diet. Strikingly, the sugar tolerant introgression line performed like parental *D. simulans*, while the sugar intolerant control line displayed higher fitness on 2.5% yeast, similar to the parental *D. sechellia* larvae (***Figure 9B***). This implies that the genetic loci of *D. simulans*, which provide high sugar tolerance cause a disadvantage in conditions of low-energy diet.

Given the observed tradeoff in sugar tolerance and starvation tolerance, we next tested the *Drosophila melanogaster* RNAi lines with sugar-intolerant phenotype for their performance on low-energy diet. Interestingly, *PPP1R15* knockdown animals showed elevated pupariation on low-energy diet when compared to corresponding control animals (***Figure 9C***). This implies that genetic changes affecting individual regulatory genes can contribute to the optimal macronutrient space of the animal.

## Discussion

In this study, we show that the macronutrient space of two closely related species that have different dietary choices is dissociated in concordance with their natural diets. Larvae of the dietary specialist *D. sechellia* that feed on a low-sugar diet in nature exhibited intolerance of high-sugar diet. In contrast, the dietary generalist *D. simulans* broadly tolerated dietary sugar, but performed poorly on low-energy-content diets. Sugar intolerance was rescued in F1 hybrids suggesting complementation of *D. sechellia* alleles with those of *D. simulans*. To identify the genomic regions associated with sugar tolerance, we introgressed a sugar tolerant phenotype into a mostly *D. sechellia* genomic background through multiple rounds of backcrossing and selection on a high-sugar diet. The sugar selected fly lines exhibited sugar tolerance equal to that of the *D. simulans* parent while the introgression control lines that were not selected on high-sugar diet exhibited very poor sugar tolerance

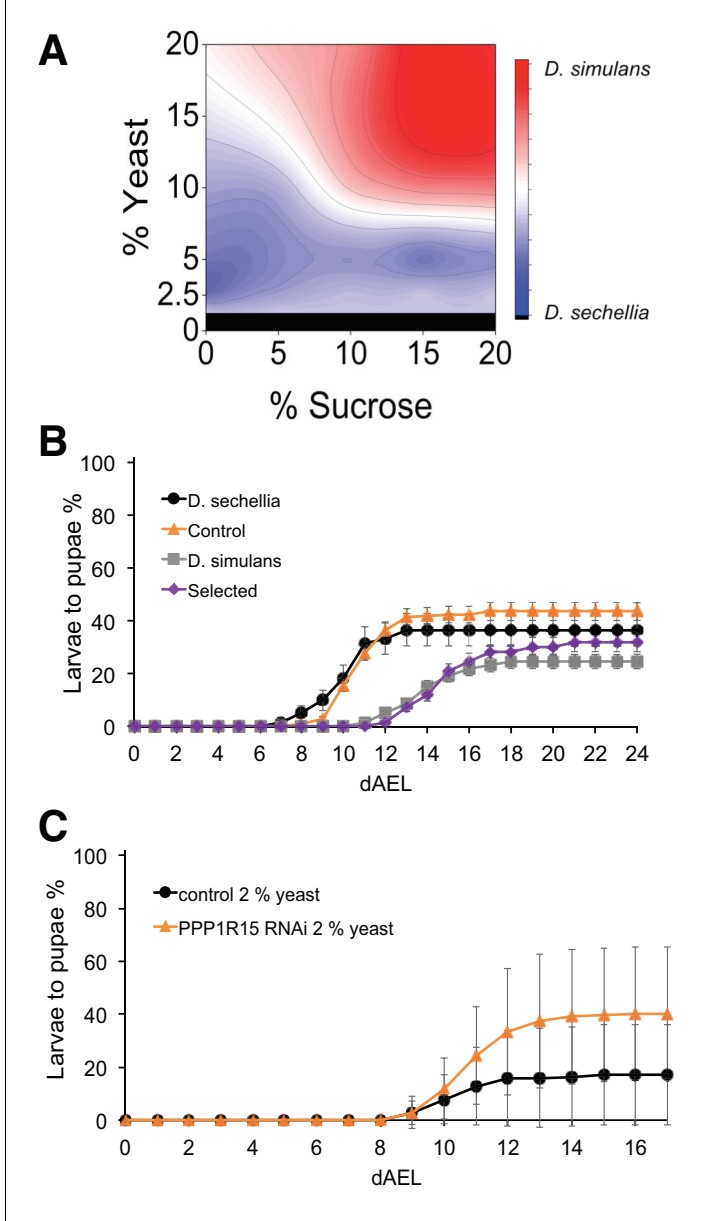

**Figure 9.** Trade-off between sugar tolerance and growth on low-energy diet. (**A**) Compared to *D. simulans*, *D. sechellia* larvae had lower tolerance of sugar, but showed an advantage in pupariation on low nutrient diets. Surface shows |(*D. sechellia* pupind) - (*D. simulans* pupind)|. (**B**) On a low nutrient (2.5% yeast) diet, *D. sechellia* and the sugar intolerant control lines had shorter egg to pupa time and greater larval survival than did *D. simulans* and the sugar-selected lines. Error bars display standard error of the mean. n = 5–9 replicates of 30 larvae/replicate for each genotype and diet. (**C**) Pupariation kinetics of *PPP1R15* RNAi (cg-GAL4>) on 2% yeast diet, n = 28 replicates of 30 larvae/replicate for each genotype and diet. Error bars display standard deviation. dAEL: days after egg laying.

DOI: https://doi.org/10.7554/eLife.40841.027

The following source data and figure supplement are available for figure 9:

**Source data 1.** Source data for *Figure 9B*.
DOI: https://doi.org/10.7554/eLife.40841.029
**Source data 2.** Source data for *Figure 9C*.
DOI: https://doi.org/10.7554/eLife.40841.030
**Figure supplement 1.** Morinda toxin tolerance is not associated with sugar tolerance.
DOI: https://doi.org/10.7554/eLife.40841.028

that was only slightly better than the *D. sechellia* parent. It remains a possibility that the dietary composition affects the growth of commensal microbes, which may differentially affect the growth of *Drosophila* species. However, we observed impaired clearance of circulating glucose in the sugar intolerant lines as well as differential gene expression response to sugar feeding. These data, together with our loss-of-function phenotypes on metabolic regulators, strongly suggests that differences in the regulation of energy metabolism were the primary causes for the observed differences in sugar tolerance. It should also be noted that our study relied on the use of single representative lines for *D. simulans* and *D. sechellia* and future studies with a larger number of lines are needed to determine the degree of natural variation of sugar tolerance within the species.

Our study represents evidence for rapid (~0.4 MY) evolution of macronutrient space in a multicellular animal. Evolution of metabolism is known to occur through multiple mechanisms, such as nonsynonymous coding region mutation, copy number variation or mutation of regulatory regions of a gene encoding a metabolic enzyme (*Wagner, 2012*). Examples of recent evolution of animal metabolism by altered function of a single enzyme include the lactase persistence in human populations (*Gerbault et al., 2011*) as well as increase in copy number of amylase-encoding gene upon dog domestication (*Axelsson et al., 2013*). In contrast to the aforementioned examples, *D. simulans* and *D. sechellia* display deviation of the macronutrient spaces along the carbohydrate/protein axis, likely requiring much more widespread reprogramming of core metabolic pathways. In line with this prediction, our RNAseq analysis revealed global changes in carbohydrate and amino acid metabolism, mitochondrial function, ribosome biogenesis and stress response pathways associated with sugar tolerance. In conclusion, our data demonstrates that global changes in macronutrient space caused by global rewiring of core metabolic pathways can occur in animals in relatively short evolutionary timeframe.

In order to reprogram large metabolic networks through mutations of genes encoding metabolic enzymes, a number of independent mutations would need to occur simultaneously, which is unlikely to occur. A plausible model for obtaining such rapid global changes in metabolic pathways is through genetic changes in metabolic 'hub' genes, including mitochondrial genes and signaling pathway components, whose activity is reflected to multiple metabolic pathways simultaneously. Several genes involved in mitochondrial ribosome were included into the introgression regions associated with sugar tolerance. The importance of mitochondrial ribosome biogenesis in survival on carbohydrate-rich food has been observed earlier (*Kemppainen et al., 2016*). Furthermore, reduced mitochondrial ribosome biogenesis is widely reflected to central carbon metabolism and redox balance of the animal (*Kemppainen et al., 2016*), which is consistent with the global gene expression differences observed in our sugar tolerant *vs.* intolerant lines.

One of the identified genetic determinants of sugar tolerance was *SERCA*, an ATP-dependent $Ca^{2+}$ pump in the ER membrane. We observed that *SERCA* displayed significantly reduced gene expression in sugar intolerant lines and that RNAi-mediated knockdown of *SERCA* caused significant sugar intolerance *D. melanogaster*. Sequence comparison of genomic regions of *D. simulans* and *sechellia* led to identification of a high level of sequence variation at the promoter, which was sufficient to explain the lower expression of *SERCA* in *D. sechellia*, based on the in vivo reporter experiment. Previous evidence shows that SERCA has a critical role in metabolic control. In *Drosophila*, *SERCA* mutant fat body cells contain reduced number and size of lipid droplets compared to wild-type (*Bi et al., 2014*). SERCA2b expression is strongly downregulated in livers of obese mice and restoring its expression is sufficient to improve glucose tolerance (*Park et al., 2010*). Similar beneficial effects have been obtained by pharmacological activation of SERCA in *ob/ob* mice (*Kang et al., 2016*). Furthermore, mice mutant for sarcolipin, a muscle-specific regulator of SERCA, are obese and have poor glucose tolerance. Thus, regulation of *SERCA* expression and activity has a significant and conserved role in the control of glucose metabolism. While it remains unclear how intracellular calcium homeostasis mechanistically regulates energy metabolism, it has been proposed that $Ca^{2+}$ transport from ER to mitochondria plays a key role (*Kaufman and Malhotra, 2014*). Interestingly, regulation of SERCA activity appears to be involved in human evolution as well. SNPs in the gene *THADA*, which encodes a regulator of SERCA activity, are among the most strongly positively selected SNPs during the evolution of modern humans, based on comparative analyses with the Neanderthal genome (*Green et al., 2010*). THADA interacts with SERCA and acts as a SERCA uncoupling protein, controlling lipid homeostasis and feeding as well as cold resistance in *Drosophila* (*Moraru et al., 2017*). In human, there is further evidence of *THADA* selection upon adaptation to

cold climate (*Cardona et al., 2014*). Future studies should explore further the role of *SERCA* and its regulators in other evolutionary processes associated with global changes in energy metabolism. Moreover, considering the role of *SERCA* in thermogenesis, it will be interesting to test, whether the lower *SERCA* expression in *D. sechellia* affects its cold tolerance.

Another interesting candidate gene identified in our study was *PPP1R15*, which had an impact on both sugar tolerance as well as survival on low-energy diet in *D. melanogaster*. Moreover, our data show that the *PPP1R15* coding region contains several amino acid changing nucleotide changes, displaying a significantly higher number than expected when compared with the other candidate genes identified ($\chi^2$ = 17.03, p < 0.01, d.f. = 5). This indicates that there has been reduced pressure of purifying selection on the gene in *D. sechellia* compared to *D. simulans*, possibly reflecting a new habitat with reduced need to maintain sugar tolerance. However, since PPP1R15 affects both sugar tolerance and starvation resistance and these two traits form a trade-off, it is also possible that the pressure for development on low-energy diet has favored an alternative form of PPP1R15, explaining the high degree of amino acid changing mutations. Functionally *PPP1R15* is an excellent candidate for a gene that enables rapid evolution of macronutrient space, since it controls major metabolic and energy consuming processes. PPP1R15 is a regulatory subunit of protein phosphatase one and it controls translation by dephosphorylating Ser51 of eIF2alpha (*Novoa et al., 2001*). eIF2alpha Ser51 is the target of the so-called integrated stress response pathway, including for example GCN2, the sensor of amino acid deprivation as well as PERK, a sensor of ER stress (*Harding et al., 1999*; *Harding et al., 2000*). It should be noted that SERCA-dependent ER $Ca^{2+}$ homeostasis has a critical role in counteracting ER stress (*Park et al., 2010*; *Lai et al., 2017*), providing a possible functional link between PPP1R15A and SERCA. Downstream of eIF2alpha (and thus PPP1R15) is transcription factor Atf4, which controls carbohydrate metabolism in *Drosophila melanogaster* (*Seo et al., 2009*; *Lee et al., 2015*). Moreover, PPP1R15A-deficient mice develop obesity, nonalcoholic liver disease, insulin resistance and impaired glucose tolerance (*Nishio and Isobe, 2015*).

In conclusion, our study provides evidence for natural variation of organismal sugar tolerance and its association with diet choice. Furthermore, our data demonstrate that global changes in metabolic gene expression, substantially affecting macronutrient space can occur in relatively short evolutionary timeframe. Our findings indicate that adaptation to a new metabolic environment, such as one with low-level nutrition, may be broadly reflected to the macronutrient space, for example by lowering the tolerance to sugar overload. This may be conceptually relevant to human health, considering that human populations with distinct histories of diet choice may bear differential vulnerabilities to nutrient overload posed by modern lifestyles.

# Materials and methods

**Key resources table**

| Reagent type (species) or resource | Designation | Source or reference | Identifiers | Additional information |
|---|---|---|---|---|
| Gene (*Drosophila melanogaster*) | mRPL43 | NA | FLYB: FBgn0034893 | |
| Gene (*D. melanogaster*) | CG4882 | NA | FLYB: FBgn0025336 | |
| Gene (*D. melanogaster*) | bonsai | NA | FLYB: FBgn0026261 | |
| Gene (*D. melanogaster*) | SERCA | NA | FLYB: FBgn0263006 | |
| Gene (*D. melanogaster*) | PPP1R15 | NA | FLYB: FBgn0034948 | |
| Gene (*D. melanogaster*) | Pi3K59F | NA | FLYB: FBgn0015277 | |

*Continued on next page*

*Continued*

| Reagent type (species) or resource | Designation | Source or reference | Identifiers | Additional information |
|---|---|---|---|---|
| Strain, strain background (*D. simulans* C167.4) | *D. Simulans* C167.4 | The National Drosophila Species Stock Center, College of Agriculture and Life Science, Cornell University | *Drosophila* Species Stock Center: 14021-0251-199 | |
| Strain, strain background (*D. sechellia* SynA) | *D. sechellia* SynA | The National Drosophila Species Stock Center, College of Agriculture and Life Science, Cornell University | *Drosophila* Species Stock Center: 14021-0248-28 | |
| Strain, strain background (*D. melanogaster* Oregon R) | *D. melanogaster* | other | | A gift from Tapio Heino, University of Helsinki |
| Strain, strain background (*D. Simulans* and *D. Sechellia* hybrid) | B.C. selected line | this paper | | See Materials and methods section 'Genetic introgression' |
| Strain, strain background (*D. Simulans* and *D. Sechellia* hybrid) | B.C. control line | this paper | | See Materials and methods section 'Genetic introgression' |
| Genetic reagent (*D. melanogaster*) | kk control | Vienna Drosophila RNAi Center | VDRC: 60100 | Genotype: y,w[1118];P{attP, y[+],w[3'] |
| Genetic reagent (*D. melanogaster*) | mRPL43 RNAi (VDRC 104466) | Vienna Drosophila RNAi Center | VDRC: 104466; FLYB: FBgn0034893 | FlyBase symbol: P{KK109027} VIE-260B |
| Genetic reagent (*D. melanogaster*) | CG4882 RNAi (VDRC 106629) | Vienna Drosophila RNAi Center | VDRC: 106629; FLYB: FBgn0025336 | FlyBase symbol: P{KK107150} VIE-260B |
| Genetic reagent (*D. melanogaster*) | bonsai RNAi (VDRC 104412) | Vienna Drosophila RNAi Center | VDRC: 104412; FLYB: FBgn0026261 | FlyBase symbol: P{KK108 444}VIE-260B |
| Genetic reagent (*D. melanogaster*) | SERCA RNAi (VDRC 107446) | Vienna Drosophila RNAi Center | VDRC: 107446; FLYB: FBgn0263006 | FlyBase symbol: P{KK107371} VIE-260B |
| Genetic reagent (*D. melanogaster*) | PPP1R15 RNAi (VDRC 107545) | Vienna Drosophila RNAi Center | VDRC: 107545; FLYB: FBgn0034948 | FlyBase symbol: P{KK104106} VIE-260B |
| Genetic reagent (*D. melanogaster*) | Pi3K59F RNAi (VDRC 100296) | Vienna Drosophila RNAi Center | VDRC: 100296; FLYB: FBgn0015277 | FlyBase symbol: P{KK107602} VIE-260B |
| Genetic reagent (*D. melanogaster*) | ubi-Gal4 | Bloomington Drosophila Stock Center | BDSC: 32551; FLYB: FBst0032551 | FlyBase symbol: w*; P{Ubi-GAL4. U}2/CyO |

*Continued on next page*

*Continued*

| Reagent type (species) or resource | Designation | Source or reference | Identifiers | Additional information |
|---|---|---|---|---|
| Genetic reagent (*D. melanogaster*) | tub-Gal4 | Bloomington Drosophila Stock Center | BDSC: 5138; FLYB: FBst0005138 | FlyBase symbol: y1 w*; P{tubP-GAL4} LL7/TM3, Sb1 Ser1 |
| Genetic reagent (*D. melanogaster*) | cg-Gal4 | Bloomington Drosophila Stock Center | BDSC: 7011; FLYB: FBst0007011 | FlyBase symbol: w1118; P{Cg-GAL4.A}2 |
| Genetic reagent (*D. melanogaster*) | fb-Gal4 | PMID: 12676093 | FLYB: FBti0013267 | Genotype: P{GAL4}fat |
| Genetic reagent (*D. melanogaster*) | *D. simulans* SERCA lacZ reporter | this paper | | Progenitors: *D. simulans* SERCA lacZ plasmid; *D. melanogaster* with landing site attP2(3L)68A4 (GenetiVision) |
| Genetic reagent (*D. melanogaster*) | *D. sechellia* SERCA lacZ reporter | this paper | | Progenitors: *D. sechellia* SERCA lacZ plasmid; *D. melanogaster* with landing site attP2(3L)68A4 (GenetiVision) |
| Recombinant DNA reagent | placZ-2.attB (vector) | PMID: 23637332 | | |
| Recombinant DNA reagent | *D. simulans* SERCA lacZ plasmid | this paper | | Progenitors: PCR, *D. simulans* C167.4 flies; vector placZ-2.attB |
| Recombinant DNA reagent | *D. sechellia* SERCA lacZ plasmid | this paper | | Progenitors: PCR, *D. sechellia* SynA flies; vector placZ-2.attB |
| Sequence-based reagent | *SERCA* F (primer) | this paper | | Sequence: 5'-TAAGCGGCCG CTCTTCGTTCAGTG GCCTGTG-3' |
| Sequence-based reagent | *SERCA* R (primer) | this paper | | Sequence: 5'-TAACTCGAG TCGTGATAAGGATT TCAGTTCG-3' |
| Sequence-based reagent | LacZ F (primer) | this paper | | Sequence: 5'-CGAATCTCTA TCGTGCGGTG-3' |
| Sequence-based reagent | LacZ R (primer) | this paper | | Sequence: 5'-CCGTTCAGCA GCAGCAGAC-3' |
| Sequence-based reagent | Act42A F (primer) | PMID: 23593032 | | Sequence: 5'-CCGTACCACAG GTATCGTGTTG-3' |
| Sequence-based reagent | Act42A R (primer) | PMID: 23593032 | | Sequence: 5'-GTCGGTTAAATC GCGACCG-3' |
| Commercial assay or kit | Glucose Oxidase/ Peroxidase assay kit (Sigma) | Sigma | Sigma: GAGO20-1KT | |

*Continued on next page*

*Continued*

| Reagent type (species) or resource | Designation | Source or reference | Identifiers | Additional information |
|---|---|---|---|---|
| Commercial assay or kit | PureGene DNA extraction kit (Qiagen) | Qiagen | Qiagen: 158667 | |
| Commercial assay or kit | Nucleospin RNA II kit (Macherey-Nagel) | Macherey-Nagel | | |
| Commercial assay or kit | SensiFast cDNA Synthesis kit (Bioline) | Bioline | | |
| Commercial assay or kit | SensiFAST SYBR No-ROX kit (Bioline) | Bioline | | |
| Software, algorithm | JMP | SAS Institute, Cary, NC | RRID:SCR_014242 | |
| Software, algorithm | PSI-seq method | PMID: 21940681 | | |
| Software, algorithm | FASTQC (v.0.11.2) | | RRID:SCR_014583 | |
| Software, algorithm | Trimmomatic (v.0.33) | | RRID:SCR_011848 | |
| Software, algorithm | Tophat (v.2.1.0) | | RRID:SCR_013035 | |
| Software, algorithm | HTseq (v.2.7.6) | | RRID:SCR_005514 | |
| Software, algorithm | R/Bioconductor package limma | | RRID:SCR_010943 | |
| Software, algorithm | R/Bioconductor package pvclust | | | URL: https://CRAN.R-project.org/package=pvclust |
| Software, algorithm | BWA-MEM (Burrows-Wheeler alignment software package) | PMID: 19451168 | | |
| Software, algorithm | Geneious 11.1.5 software | Biomatters Ltd., Aukland, NZ | RRID:SCR_010519 | |

## Fly food, fly stocks and husbandry

*Drosophila simulans* line C167.4 (14021-0251-199) and *D. sechellia* line SynA (14021-0248-28) were obtained from the *Drosophila* Species Stock Center, University of California, San Diego (now located at the College of Agriculture and Life Science, Cornell University). *D. melanogaster* strain Oregon R was a gift from Tapio Heino, University of Helsinki. The following VDRC RNAi lines were used: *mRpL43*: 104466, *CG4882*: 106629, *bonsai*: 104412, *SERCA*: 107446, *PPP1R15* (*Gadd34*): 107545, *Pi3K59F*: 100296, *Taldo*: 106308, *Dpit47*: 110401, *GlcT-1*: 108064 (see VDRC web site for specific information). *Ubi-GAL4*, *tub-GAL4* and *cg-GAL4* were obtained from the Bloomington *Drosophila* Stock Center. *Fb-GAL4* (FBti0013267 (*Grönke et al., 2003*) was also used. All fly stocks and parents of the experimental flies were maintained on a common laboratory diet containing 2.4% (v/v) nipagin, 0.7% (v/v) propionic acid. All experiments took place at 25℃, 50% RH with 12 hr light, 12 hr dark daily cycle and at controlled density (30 larvae *per* vial). Driver lines crossed with $w^{1118}$ containing landing site VIE-260B (VDRC ID: 60100) were used as controls for the RNAi experiments.

Different driver lines were used in the knockdown, depending on the strength of the phenotypes. For selection on *Morinda* toxins flies were reared on 0.5% agar 20% yeast diet supplemented with 0.5% hexanoic acid and 0.01% octanoic acid (*Earley and Jones, 2011*).

## Nutrient space pupariation assay

We determined the optimal nutrient space for larval growth for each species. Parents of the experimental flies were released into egg-laying chambers provided with apple-juice-agar plates supplemented with yeast and allowed to lay eggs for 2 hr intervals. Yeast was removed from the egg-laying plates and they were incubated 24 hr at 25°C, 50% RH. Thirty 1st instar larvae were placed into five replicate vials of 0.5% agar-based media in a 5 × 5 grid of baker's yeast (1.25, 2.5, 5, 10 and 20%) and sucrose (0, 5, 10, 15, and 20%). Estimated caloric contents of the diets are presented in *Table 5*. Vials were scored for the number of larvae pupariated at 24 hr intervals for 408 hr total. Because nutrition affects both developmental timing and larval survival, we calculated the pupariation index (Pupind) that is the maximum rate of pupariation. Specifically, $Pupind = max(\{(p_t/t): t = 24, 48, \ldots, 408\})$; where $p_t$ = number of larvae that have pupariated at observation time $t$ hours after egg-laying (hAEL). The maximum Pupind was analyzed using a full general linearized model with main effects of genotype, yeast, sucrose and their interactions using JMP software (SAS Institute). The strengths of the main effects were determined by calculating $\omega^2$ (*Yigit and Mendes, 2018*).

## Feeding behavior

Feeding behavior was assayed by quantifying the rate of larval mouth hook extensions using the method described by *Shen (2012)*. Parents of the experimental larvae were allowed to lay eggs for 2 hr on apple juice agar plates spread with yeast paste. Plates were incubated at 25°C overnight and then first instar larvae were transferred to vials of 20% yeast in groups of 30 and raised to pre-wandering third larval instar. On the morning of the assay, larvae were transferred to Petri plates containing 20% yeast colored with blue food dye and were allowed to feed for 1 hr. Actively feeding larvae with visible blue food in the gut were selected for assay.

Media for the mouth hook extension assay was prepared by mixing 12 g dry agar with a solution of 1 × PBS buffer or with 1 × PBS buffer containing 20% sucrose to final volume of 100 ml. The mixtures were incubated overnight at 4°C to hydrate the agar completely. The thickened assay media was poured into small Petri plates and allowed to equilibrate to RT for 3 hr.

To begin the assay, 30 larvae were transferred from the blue-dyed LSD plate onto a plate of assay media. Larvae were viewed on the plate using a dissecting microscope and the number of mouth hook extensions in 1 min was counted for 10 individual larvae. Larvae float on the assay media and are unable to move from where they are placed. Four no sucrose and four 20% sucrose plates were observed. Counts were recorded using a handheld cell counter and each observed larva was removed from the plate before proceeding to the next. Mean mouth hook extension rate was calculated per plate from the 10 larvae and compared by ANOVA with species and sucrose percent as main effects.

**Table 5.** Estimated caloric content of the 25 Yeast-Sugar diets (kcal/100 g).

|  |  | % Yeast | | | | |
|---|---|---|---|---|---|---|
|  |  | 1.25 | 2.5 | 5 | 10 | 20 |
| % Sugar | 0 | 4.1 | 8.1 | 16.3 | 32.5 | 65.0 |
|  | 5 | 24.4 | 28.4 | 36.6 | 52.8 | 85.3 |
|  | 10 | 44.7 | 48.7 | 56.9 | 73.1 | 105.6 |
|  | 15 | 65.0 | 69.0 | 77.2 | 93.4 | 125.9 |
|  | 20 | 85.3 | 89.3 | 97.5 | 113.7 | 146.2 |

DOI: https://doi.org/10.7554/eLife.40841.031

## Phenotype-based selection and introgression with resequencing

We introgressed the phenotype of sugar tolerance from *D. simulans* into a mostly *D. sechellia* genomic background using the crossing scheme outlined in *Figure 2A*. To produce F1 hybrids, groups of 100 unmated *D. simulans* females were collected and crossed to 100 *D. sechellia* males in 1-litre plastic population cages. Females were allowed to lay eggs on apple-juice agar plates for 24 hr, then eggs were collected and seeded at approximately 200 eggs per 240 ml bottle onto 20% yeast media (*Clancy and Kennington, 2001*). For the first backcross generation $\geq$100 unmated F1 hybrid females were collected from the bottles and crossed to 100 *D. sechellia* males in 1-liter plastic population cages. Females from the cross were allowed to lay eggs on apple-juice agar plates for 24 hr then eggs were collected and seeded at approximately 200 eggs per 240 ml bottle onto 20% yeast/ 20% sucrose media to select the sugar tolerant phenotype or onto 20% yeast for the control backcross. Bottles of eggs that were collected from a single crossing cage were kept together, separate from those collected from replicate crosses. For generation 2 through 10, virgin females were collected from the bottles, crossed to *D. sechellia* males, eggs were collected and seeded into fresh 20% yeast/20% sucrose (selected) or 20% yeast (control) media. Crossing populations consisted of $\geq$100 backcross females from the previous generation and 100 *D. sechellia* males. After 10 generations of backcrossing the lines were sibling mated for three generations before beginning experiments. Backcross lines were continuously maintained on 20% yeast/20% sucrose (selected) or 20% yeast (control) diet throughout sibling mating.

Genomic DNA was extracted from 30 female flies for two no-selection control lines, three sugar-selected lines, and from each parental *D. sechellia* and *D. simulans* line using a PureGene DNA extraction kit (Qiagen). Resequencing was performed at the University of North Carolina DNA sequencing facility. Regions of introgression were mapped using the PSI-seq method of *Earley and Jones (2011)*.

Original datasets have been placed into a public repository (NCBI): https://www.ncbi.nlm.nih.gov/bioproject/PRJNA486014/

## Hemolymph glucose clearance

We determined the hemolymph glucose concentration of the parental and introgression lines after feeding on a high sugar diet. Larvae were collected onto 20% yeast media and raised to pre-wandering third larval instar. Five replicates of 50 pre-wandering third instar larvae were transferred to media containing 20% sucrose for 2 hr and then to 0% sucrose media for 2 hr. Hemolymph was collected from 10 larvae at time points 0 hr - 0%, 2 hr - 20% sucrose and 2 hr - clearance (0% sucrose). Hemolymph glucose was assayed using a Glucose Oxidase/Peroxidase assay kit (Sigma) (*Havula et al., 2013*) and data were analyzed by comparing to *D. simulans* using Dunnett's test implemented in JMP software (SAS institute). Dunnett's test queries whether the difference between the mean of the control group and an experimental group differs by greater than a critical value.

## RNA sequencing and data analysis

We extracted and sequenced total RNA from third instar *D. simulans* (C167.4), *D. sechellia* (SynA), backcross control, and backcross selected larvae that were fed LSD (20% yeast) or HSD (20% yeast/ 20% sucrose) for 8 hr using a Nucleospin RNA II kit (Macherey-Nagel). Larvae were prepared from three replicates of each fly line. Parents of the experimental larvae were released into egg collection cages and allowed to lay eggs for 2 hr on apple juice agar plates supplemented with yeast paste. Plates of eggs were placed at 25°C for 24 hr. Following incubation, 100 first instar larvae were transferred to plates containing LSD and placed at 25°C for 48 hr after which pre-wandering third instar larvae were transferred to plates of LSD or HSD that contained 2% blue food dye. Larvae were allowed to feed for 2 hr at 25°C then larvae that did not have blue dye in their gut were removed from the plates. Feeding larvae that had blue dye in their gut were kept 6 hr at 25°C and then collected for RNA extraction. RNA sequencing libraries we prepared using the TruSeq Stranded mRNA Library Prep Kit (Illumina) and sequenced (single-end 76 bp reads) using Illumina NextSeq 500 technology.

The quality of the reads was assessed with FASTQC (v.0.11.2) (Babraham Bioinformatics, Cambridge, UK). The reads were trimmed with Trimmomatic (v.0.33) (*Bolger et al., 2014*). Reads were scanned with sliding window of 20 and if the average quality per base dropped below 20, the read

was discarded. Additionally, base length of 40 was required for the reads and for both leading and trailing ends quality score of 30 was required. For the *mlx* mutant RNAseq dataset (*Mattila et al., 2015*), trimming was performed with sliding window of 4 bases with average per base quality requirement of 15. Required base length was set to 36 and required strand quality in both at the end and start was set to 36. Tophat (v.2.1.0) (*Trapnell et al., 2009*) was used for aligning the reads to *D. melanogaster* reference genome (Flybase R6.10). HTseq (v.2.7.6) (*Anders et al., 2015*) was used for strand-specific quantification of exons. Reads with quality score below 10 were discarded.

The quality of the samples was assessed with multi-dimensional scaling and variety of sample clustering methods (pearson correlation, euclidean ward, euclidean complete) using R/Bioconductor's package pvclust (*Suzuki and Shimodaira, 2006*). Based on these results, one sample (sim.0.1) was defined as outlier and thus removed from the analysis. The differential expression analysis was performed with R/Bioconductor package limma (*Ritchie et al., 2015*; *Law et al., 2014*). The samples were required to have >1 CPM (counts per million) in all replicates in either tolerant or intolerant group. For *mlx* mutant datasets, no outliers were discovered, and >1 CPM was required in majority of samples in at least one of the conditions. The Benjamini-Hochberg correction was used for adjusting p values (*Benjamini and Hochberg, 1995*).

Sample clustering was performed using R/Bioconductor package pvclust (*Suzuki and Shimodaira, 2006*). Correlation was used as distance matrix. The gene set enrichment was performed with hypergeometric test for the manually downloaded pathway sets from KEGG and GO. The pathway was defined as enriched if the adjusted p value < 0.05. The heatmaps were generated using scaled log2CPM values for means of each sample group. The scaling was performed separately for the two datasets. The row-wise clustering was performed using correlation distance.

Original datasets have been placed into a public repository (NCBI): https://www.ncbi.nlm.nih.gov/bioproject/PRJNA486014/.

## Comparative analysis of genomic regions of candidate genes

Species-specific nucleotide sequences for *SERCA, CG4882*, *RpL49,* and *bonsai* genomic regions were compiled by mapping Illumina sequence reads from *D. simulans* (C167.4) and *D. sechellia* (SynA) to *D. melanogaster* sequences using BWA-MEM in the Burrows-Wheeler alignment software package (*Li and Durbin, 2009*). Sequence read alignments were edited by hand using Geneious 11.1.5 software (Biomatters Ltd., Aukland, NZ) to produce simple majority consensus sequences. For each gene, the consensus sequences were aligned and nucleotide differences between species called using Geneious 11.1.5 software. The average frequency of nucleotide differences was calculated in 100 base windows slid forward in steps of 25 bases. The genomic average frequency of nucleotide differences between *D. simulans* and *D. sechellia* was calculated for three randomly chosen 20 kb regions on chromosomes 2R, 2L, and X, and was subtracted from each window to correct for background noise. Frequencies were charted using JMP Pro 14 software (SAS Institute Cary, NC).

## Generation of transgenic reporter flies

The likely *SERCA* promoter region was identified based on the nine chromatin stage model (*Kharchenko et al., 2011*), the selected fragment corresponds to the 'red' chromatin type (Active promoter/transcription start site region). The selected 1.2 kB promoter regions of *SERCA* from *D. simulans* (C167.4) and *D. sechellia* (SynA) were cloned into the placZ-2.attB vector using restriction enzyme sites *Not*I and *Xho*I and ligase-dependent cloning (*Bischof et al., 2013*). Successful generation of plasmids was verified with Sanger sequencing. Injection was performed by GenetiVision (Houston, TX) into a *D. melanogaster* w[1118] line with landing site attP2(3L)68A4.

Cloning was performed using the following primers:
*SERCA* F:
5'-TAAGCGGCCGCTCTTCGTTCAGTGGCCTGTG-3'
*SERCA* R:
5'-TAACTCGAGTCGTGATAAGGATTTCAGTTCG-3'

## RNA extraction and quantitative RT-PCR

Eight early 3rd instar larvae were collected for each sample and RNA was extracted using the Nucleospin RNA kit (Macherey-Nagel) according to the manufacturer's protocol. RNA was reverse-transcribed using the SensiFAST cDNA Synthesis kit (Bioline) according to the manufacturer's protocol, and qPCR was performed using SensiFAST SYBR No-ROX kit (Bioline) with Light cycler 480 Real-Time PCR System (Roche) with three technical replicates per sample. Actin 42A was used as a reference gene.

Following primers were used:
LacZ F: 5'-CGAATCTCTATCGTGCGGTG-3'
LacZ R: 5'-CCGTTCAGCAGCAGCAGAC-3'
Act42A F: 5'-CCGTACCACAGGTATCGTGTTG-3'
Act42A R: 5'-GTCGGTTAAATCGCGACCG-3'

## Acknowledgements

We thank Christen Mirth, Ronald Regal, and members of our group for advice and comments. Sequencing was performed by the core facility (DNA Sequencing and Genomics Laboratory) of the Institute of Biotechnology, University of Helsinki. Funding was provided by the Academy of Finland (grant no. 286767 to VH), Novo Nordisk Foundation (NNF16OC0021460 to VH), Sigrid Juselius Foundation and Finnish Diabetes Foundation.

## Additional information

### Funding

| Funder | Grant reference number | Author |
| --- | --- | --- |
| Suomen Akatemia | 286767 | Ville Hietakangas |
| Novo Nordisk Fonden | NNF16OC0021460 | Ville Hietakangas |
| Sigrid Juséliuksen Säätiö | | Ville Hietakangas |
| Diabetestutkimussäätiö | | Ville Hietakangas |

The funders had no role in study design, data collection and interpretation, or the decision to submit the work for publication.

### Author contributions

Richard G Melvin, Conceptualization, Formal analysis, Supervision, Funding acquisition, Investigation, Visualization, Writing—original draft, Project administration; Nicole Lamichane, Conceptualization, Formal analysis, Investigation, Visualization, Writing—original draft, Writing—review and editing; Essi Havula, Formal analysis, Investigation, Visualization, Writing—review and editing; Krista Kokki, Formal analysis, Visualization, Writing—review and editing; Charles Soeder, Formal analysis, Writing—review and editing; Corbin D Jones, Resources, Methodology, Writing—review and editing; Ville Hietakangas, Conceptualization, Supervision, Funding acquisition, Investigation, Writing—original draft, Project administration, Writing—review and editing

### Author ORCIDs

Richard G Melvin https://orcid.org/0000-0001-6428-6763
Nicole Lamichane http://orcid.org/0000-0002-1746-0332
Ville Hietakangas https://orcid.org/0000-0002-9900-7549

### Decision letter and Author response

Decision letter https://doi.org/10.7554/eLife.40841.039
Author response https://doi.org/10.7554/eLife.40841.040

## Additional files

### Supplementary files

• Supplementary file 1. *D. melanogaster* genes corresponding to the *D. simulans/sechellia* genes in the introgressed genomic regions. Genes which were up- or downregulated in both sugar intolerant genotypes when compared to both sugar tolerant genotypes are indicated as well as the genes that were screened by RNAi.
DOI: https://doi.org/10.7554/eLife.40841.032

• Transparent reporting form
DOI: https://doi.org/10.7554/eLife.40841.033

### Data availability

Genome sequencing and RNA sequencing datasets have bee placed into NCBI SRA archive, Study # SRP158000. A link is provided for reviewers in the Materials and Methods.

The following dataset was generated:

| Author(s) | Year | Dataset title | Dataset URL | Database and Identifier |
|---|---|---|---|---|
| Mattila J, Nicole La-michane, Essi Havu-la, Krista Kokki, Charles Soeder, Corbin D Jones, Ville Hietakangas | 2018 | Natural variation in sugar tolerance | https://www.ncbi.nlm.nih.gov/bioproject/PRJNA486014/ | NCBI SRA archive, PRJNA486014 |

The following previously published dataset was used:

| Author(s) | Year | Dataset title | Dataset URL | Database and Identifier |
|---|---|---|---|---|
| Mattila J, Havula E, Suominen E, Tee-salu M | 2015 | Sugar responsive regulatory network that controls organismal carbohydrate, amino acid and lipid homeostasis | https://www.ncbi.nlm.nih.gov/geo/query/acc.cgi?acc=GSE70980 | NCBI Gene Expression Omnibus, GSE70980 |

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
