## [Decision Letter]

Thank you for submitting your article "Natural variation in sugar tolerance associates with changes in signaling and mitochondrial ribosome biogenesis" for consideration by *eLife*. Your article has been reviewed by Patricia Wittkopp as the Senior Editor, a Reviewing Editor, and three reviewers. The following individual involved in review of your submission has agreed to reveal her identity: Christen K Mirth (Reviewer #3).

The reviewers have discussed the reviews with one another and the Reviewing Editor has drafted this decision to help you prepare a revised submission.

The manuscript by Melvin and co-authors outlines a thorough and elegant study to explore the genetic nature of differences in dietary requirements between species. This study uses the Nutritional Geometry framework to explore how two closely related species, *Drosophila simulans* and *D. sechellia* differ in their tolerance for sugar. They further extend this beyond the scope of previous nutritional geometry studies to understand the metabolic differences underpinning differences in sugar tolerance, using a clever backcrossing approach to introgress *D. simulans* sugar tolerance into the *D. sechellia* genomic background. In their final experiment, they examine trade-offs between high sugar intolerance and the ability to survive on a low nutrient diet. They provide evidence that at the cost of high-sugar intolerance, *D. sechellia* survive better on low energy diets than *D. simulans*.

This is a colossal study, with far-reaching implications on how dietary requirements are determined. Each of these experiments represent an impressive amount of work, and as a reward for their efforts the authors have generated one of the most complete pictures of how differences in dietary ranges are encoded in the genome. The study is of exceptionally high quality, but the reviewers did raise a few questions about the analysis and interpretations that will need to be addressed before publication. Addressing this and the other comments shown below (which are combined from the three reviewers and thus not always in the order in which the issues appear in the manuscript) will require extensive changes to the text.

Especially important to address among these comments are the requests for (1) more specific methodological details including for the pupariation index, the experimental parameters for RNAseq libraries etc. and the comments relating to the discussion, (2) clearer aims and reworking of the document for better logical flow, and (3) revisiting the nutritional geometry analysis and presenting the results with a greater degree of caution in the Discussion section (acknowledging that because you used a single inbred line from each species, it is difficult to be certain that these are truly differences between species).

For the response surface analysis, you should be more cautionary with your interpretation. For a response surface to be considered strikingly different, the optima is expected to be at opposite ends of the nutrient space. The main differences that you see are that responses for survival, development time, and pupariation index are much more restricted in *D. sechellia* than in *D. simulans*. In other words, the optima overlap but occupy a much broader region in *D. simulans*.

To infer significant differences between responses you need to do a formal test. It's true you find significant correlations between your trait and sucrose concentrations only in *D. sechellia* (and F1 hybrids for pupariation index) but given that variance in these datasets tends to be high this does not necessarily mean that the responses are significantly different. To test this, you should include a full glm model with all interactions: trait ~ sucrose + sucrose2 + yeast + yeast2 + species + sucrose*yeast + sucrose*species + yeast*species + sucrose*yeast*species. If any of the interaction terms between diet and species are significant (sucrose*species + yeast*species + sucrose*yeast*species), you can be confident that there is a significant difference in the way each species responds to the composition of the diet. It also provides additional information as it tells you what is driving the difference; the sucrose*species term would be expected to be significant but not the yeast*species term – meaning that the differences in how traits respond to the diet are driving by differences in sucrose tolerance (but not protein requirements) between *D. simulans* and *D. sechellia*.

The rationale for comparing expression between your introgression lines to mlx mutants was not clear. Was this done to show that the intolerant lines show similar expression profiles as previously identified sugar-intolerant mutants?

When using a single inbred line for each species, it's impossible to say whether the differences observed are truly due to differences between species. For example, without knowing how *D. simulans* populations vary in sugar tolerance, it's entirely possible that the inbred line selected showed high tolerance relative to the population mean (or that the *D. sechellia* line chosen had low sugar tolerance). There are two ways around this problem, you can conduct your nutritional geometry experiments on genetically diverse outbred lines or you can repeat these experiments on multiple inbred lines of the same species (to get an estimate for genetic variation for your trait within a species). Given the size of nutritional geometry experiments, using outbred lines is usually preferred. All this to say, it is difficult to be sure from your experiments that the differences you've mapped really represent differences between species. While it is not necessary to directly address this problem within the context of this work, this caveat should be mentioned in the Discussion section. It doesn't detract from your main message: that you've explored the difference in sugar tolerance between two genomes and identified genomic regions that contribute to this variation.

The order of the Materials and methods section does not parallel the order of the Results section. In addition, the order of the methods does not follow a logical progression (ex. Hemolymph glucose clearance assay which requires the introgressed lines is presented before Genomic Introgression).

Each section within the Results section ends with a sentence that begins "In conclusion”. The authors then make a statement that discusses the results and should be incorporated in the Discussion section not the Results section.

Nowhere in the Introduction do the authors state what they aim to do with the study. They present their hypothesis but don't provide context as to how they intend to test it in the study.

The subsection “Introgression of sugar tolerance phenotype” is primarily describing methods and would be better placed in the Materials and methods section. Also, in the Materials and methods section, the authors do not provide enough detail to replicate this method.

The Abstract primarily presents results but does not offer sufficient background detail to frame the importance of the study.

---

## [Author Response]

The manuscript by Melvin and co-authors outlines a thorough and elegant study to explore the genetic nature of differences in dietary requirements between species. This study uses the Nutritional Geometry framework to explore how two closely related species, Drosophila simulans and D. sechellia differ in their tolerance for sugar. They further extend this beyond the scope of previous nutritional geometry studies to understand the metabolic differences underpinning differences in sugar tolerance, using a clever backcrossing approach to introgress D. simulans sugar tolerance into the D. sechellia genomic background. In their final experiment, they examine trade-offs between high sugar intolerance and the ability to survive on a low nutrient diet. They provide evidence that at the cost of high-sugar intolerance, D. sechellia survive better on low energy diets than D. simulans.This is a colossal study, with far-reaching implications on how dietary requirements are determined. Each of these experiments represent an impressive amount of work, and as a reward for their efforts the authors have generated one of the most complete pictures of how differences in dietary ranges are encoded in the genome. The study is of exceptionally high quality, but the reviewers did raise a few questions about the analysis and interpretations that will need to be addressed before publication. Addressing this and the other comments shown below (which are combined from the three reviewers and thus not always in the order in which the issues appear in the manuscript) will require extensive changes to the text.Especially important to address among these comments are the requests for (1) more specific methodological details including for the pupariation index, the experimental parameters for RNAseq libraries etc. and the comments relating to the discussion, (2) clearer aims and reworking of the document for better logical flow, and (3) revisiting the nutritional geometry analysis and presenting the results with a greater degree of caution in the Discussion section (acknowledging that because you used a single inbred line from each species, it is difficult to be certain that these are truly differences between species).

We thank the reviewers for a thorough and constructive feedback, which has helped us to significantly improve our study. We have now revised the manuscript according to the comments of the reviewers, including the three points highlighted above. Below, we provide a point-by-point response to the reviewers.

For the response surface analysis, you should be more cautionary with your interpretation. For a response surface to be considered strikingly different, the optima is expected to be at opposite ends of the nutrient space. The main differences that you see are that responses for survival, development time, and pupariation index are much more restricted in D. sechellia than in D. simulans. In other words, the optima overlap but occupy a much broader region in D. simulans.

We have now modified the Results section to more precisely describe the phenotypes, e.g. “In contrast, *D. sechellia* larvae displayed a more restricted space, with slowed development and reduced survival on high protein diets containing >10% sugar and complete lethality on diet composed of 20% sucrose/20% yeast (Figure 1A, Table 1).”

To infer significant differences between responses you need to do a formal test. It's true you find significant correlations between your trait and sucrose concentrations only in D. sechellia (and F1 hybrids for pupariation index) but given that variance in these datasets tends to be high this does not necessarily mean that the responses are significantly different. To test this, you should include a full glm model with all interactions: trait ~ sucrose + sucrose2 + yeast + yeast2 + species + sucrose*yeast + sucrose*species + yeast*species + sucrose*yeast*species. If any of the interaction terms between diet and species are significant (sucrose*species + yeast*species + sucrose*yeast*species), you can be confident that there is a significant difference in the way each species responds to the composition of the diet. It also provides additional information as it tells you what is driving the difference; the sucrose*species term would be expected to be significant but not the yeast*species term – meaning that the differences in how traits respond to the diet are driving by differences in sucrose tolerance (but not protein requirements) between D. simulans and D. sechellia.

We have now tested for significant differences in the responses using the full glm model with all interactions. The results of the analysis are presented in the new Table 2 and show that all effects and their interactions are significant. To get at the question of what drives differences in trait values we calculated ω^2^ to estimate the strength of the sucrose, yeast, and sucrose × yeast interaction effects for each species and the F1 hybrid separately. We find that the effect of sucrose on Pupind, survival, and development time is substantially stronger in *D. sechellia* than in *D. simulans* (Table 3). Conversely, the effect of yeast is stronger in *D. simulans* than *sechellia* (reflecting the better performance of *sechellia* in low yeast). However, the effects indicate that the species difference is much greater for sucrose, further underlining the prominent sugar intolerance in *D. sechellia*.

The rationale for comparing expression between your introgression lines to mlx mutants was not clear. Was this done to show that the intolerant lines show similar expression profiles as previously identified sugar-intolerant mutants?

We have now better clarified the rationale in the Results section.

When using a single inbred line for each species, it's impossible to say whether the differences observed are truly due to differences between species. For example, without knowing how D. simulans populations vary in sugar tolerance it's entirely possible that the inbred line selected showed high tolerance relative to the population mean (or that the D. sechellia line chosen had low sugar tolerance). There are two ways around this problem, you can conduct your nutritional geometry experiments on genetically diverse outbred lines or you can repeat these experiments on multiple inbred lines of the same species (to get an estimate for genetic variation for your trait within a species). Given the size of nutritional geometry experiments, using outbred lines is usually preferred. All this to say, it is difficult to be sure from your experiments that the differences you've mapped really represent differences between species. While it is not necessary to directly address this problem within the context of this work, this caveat should be mentioned in the Discussion section. It doesn't detract from your main message: that you've explored the difference in sugar tolerance between two genomes and identified genomic regions that contribute to this variation.

As suggested, we have now mentioned in the Discussion section the need to determine the variation of sugar tolerance within the species by using multiple lines.

The order of the Materials and methods section does not parallel the order of the Results section. In addition, the order of the methods does not follow a logical progression (ex. Hemolymph glucose clearance assay which requires the introgressed lines is presented before Genomic Introgression).

We have now reorganized the Materials and methods section, to match the order of the Results section.

Each section within the Results section ends with a sentence that begins "In conclusion”. The authors then make a statement that discusses the results and should be incorporated in the Discussion section not the Results section.

As suggested, we have omitted the concluding sentences from the Results section.

Nowhere in the Introduction do the authors state what they aim to do with the study. They present their hypothesis but don't provide context as to how they intend to test it in the study.

We have now stated the aim of the study in the Introduction: “Here we aimed to explore the natural variation of macronutrient space in closely related species.”

The subsection “Introgression of sugar tolerance phenotype” is primarily describing methods and would be better placed in the Materials and methods section. Also, in the Materials and methods section, the authors do not provide enough detail to replicate this method.

As suggested, we have reduced the methodological descriptions in the Results section and better described the details of the introgression experiment in the Materials and methods section.

The Abstract primarily presents results but does not offer sufficient background detail to frame the importance of the study.

We have now revised the Abstract to better frame the importance of the study, some details of results were omitted due to length restriction.